# Energy Storage Application of All-Organic Polymer Dielectrics: A Review

**DOI:** 10.3390/polym14061160

**Published:** 2022-03-14

**Authors:** Zhijie Yang, Dong Yue, Yuanhang Yao, Jialong Li, Qingguo Chi, Qingguo Chen, Daomin Min, Yu Feng

**Affiliations:** 1Key Laboratory of Engineering Dielectrics and Its Application, Ministry of Education, Harbin University of Science and Technology, Harbin 150080, China; yangzhijiehrbust@163.com (Z.Y.); yhyao@hrbust.edu.cn (Y.Y.); qgchi@hrbust.edu.cn (Q.C.); qgchen@263.net (Q.C.); 2School of Electrical and Electronic Engineering, Harbin University of Science and Technology, Harbin 150080, China; 3School of Materials Science and Chemical Engineering, Harbin University of Science and Technology, Harbin 150080, China; 4School of Material Science and Engineering, Shaanxi University of Science and Technology, Xi’an 710021, China; 5State Key Laboratory of Electrical Insulation and Power Equipment, Xi’an Jiaotong University, Xi’an 710049, China

**Keywords:** all-organic polymers, energy storage performance, dielectric properties

## Abstract

With the wide application of energy storage equipment in modern electronic and electrical systems, developing polymer-based dielectric capacitors with high-power density and rapid charge and discharge capabilities has become important. However, there are significant challenges in synergistic optimization of conventional polymer-based composites, specifically in terms of their breakdown and dielectric properties. As the basis of dielectrics, all-organic polymers have become a research hotspot in recent years, showing broad development prospects in the fields of dielectric and energy storage. This paper reviews the research progress of all-organic polymer dielectrics from the perspective of material preparation methods, with emphasis on strategies that enhance both dielectric and energy storage performance. By dividing all-organic polymer dielectrics into linear polymer dielectrics and nonlinear polymer dielectrics, the paper describes the effects of three structures (blending, filling, and multilayer) on the dielectric and energy storage properties of all-organic polymer dielectrics. Based on the above research progress, the energy storage applications of all-organic dielectrics are summarized and their prospects discussed.

## 1. Introduction

With the functionalization of modern power systems and power electronic devices, the development of high-power and high-energy storage capacitors has become a top priority [1,2]. Dielectric capacitors have rapid charging and discharging speeds and low density and are light in terms of weight; they are widely used in pulsed power devices in the electrical and electronic engineering fields. In comparison to inorganic ceramic capacitors, polymer-based capacitors exhibit numerous advantages, such as high breakdown strength, low dielectric loss, easy processing and production, and low cost. They also have good application prospects in hybrid electric vehicles, microelectronic systems, wind-power generation, power transmission, microwave communications, and underground oil and gas exploration (Figure 1) [3,4,5,6,7,8].

The development of high energy storage density dielectrics has become an issue that is currently being focused on. At present, the most commonly used dielectric material is biaxially oriented polypropylene (BOPP), but its low energy storage density (1–2 J/cm^3^) presents challenges for development. Moreover, the effects of temperature also negatively impact its energy storage density and efficiency [9,10,11,12,13,14,15].

In the early stages of the research study, the researchers focused on improving the dielectric capacitors’ dielectric constant (*ε*_r_) and breakdown strength (*E*_b_). However, in polymers, it is difficult to achieve both high breakdown strength and high polarization characteristics. Therefore, the following are primary problems in the development of new high-performance polymer dielectrics that require resolution: strong coupling between breakdown strength and dielectric constant from a scientific point of view, and service life from the perspective of engineering applications.

Due to different research studies on capacitor energy storage strategies, there are three main methods that improve energy storage density: The first method involves constructing filled composite dielectrics; the second method requires designing layered composite dielectrics; and the third method involves the synthesis of new polymers [16,17,18,19,20,21,22,23].

Filled composite dielectrics are created by adding various fillers, such as ceramics, metal oxides, and conductive materials, to the polymer matrix. The filling process enhances the interface polarization and increases the dielectric constant of the composites, but at the same time, it also reduces breakdown strength [24,25,26,27]. Filling undoubtedly determines the interface area, which requires smaller fillers. Thus, nano-sized fillers are more commonly used than micro-sized fillers. To increase the dielectric constant of the composites, a large amount of inorganic filler is required, which poses a problem. That is, the high inorganic metal oxide content will destroy the polymer’s good flexibility and processing properties [28,29,30,31].

Layered composite dielectrics are polymers with a sandwich structure or multilayer structure. By utilizing different functions between the layers, high dielectric constant layers and high breakdown strength layers are superimposed layer-by-layer. In the delamination effect, each layer material’s dielectric constant and conductivity are fixed. Although the multilayer interface can be used to suppress carrier migration and improve insulation performance, it does not greatly improve polarization performance [32,33,34,35,36,37].

Based on the above-mentioned information, this article reviews and summarizes all-organic composite materials. Polymer-based dielectrics are divided into linear polymer dielectrics and nonlinear polymer dielectrics. Their advantages and potential are described in terms of three structures, namely: filling, blending, and multilayer. Lastly, we describe the future development prospects and challenges of all-organic composite materials.

## 2. Energy Storage Characteristic Parameters

### 2.1. Charge-Discharge Energy Density

The capacitor is composed of parallel plates and dielectric materials. When an electric field is applied, the dielectric in the plate will be polarized. The energy storage capacity of a capacitor can be quantified by capacitance *C*, which is defined as [4]:(1)$$C=\varepsilon_{0}\varepsilon_{r}\frac{A}{d}$$
where *ε*_0_ and *ε*_r_ are the vacuum dielectric constant and the relative dielectric constant, while A and d are the area of the dielectric’s plates and the thickness between the plates, respectively. 

Figure 2 shows the typical charge-discharge cycle process of the dielectric, in which the sum of the blue and green areas is the charged energy density. In general, the dielectric energy storage density formula is detailed as follows [38]:(2)$$U_{\mathrm{storage}}=\frac{W}{\mathrm{Ad}}=\frac{\int_{0}^{Q_{\max}}VdQ}{\mathrm{Ad}}=\int_{0}^{D_{\max}}EdD$$
where W represents energy storage, and U_storage_ is defined as energy storage density. *Q*/A shows that charge density is expressed by electric displacement (*D*), where *E* (V/d) is the applied electric field. 

In terms of linear dielectrics, the dielectric constant has nothing to do with the applied electric field. The relationship between the dielectric constant, the electric field, and energy storage density is shown in the following formula [23]:(3)$$U_{\mathrm{storage}}=\int_{0}^{D_{\max}}EdD=\int_{0}^{E_{\max}}EdD=\frac{1}{2}\varepsilon_{0}\varepsilon_{r}E^{2}$$

Energy storage density is proportional to the square of the applied electric field. However, this formula is not suitable for nonlinear dielectrics, as the dielectric constant of nonlinear dielectrics will change alongside the external electric field.

For nonlinear dielectrics, such as ferroelectrics, relaxor ferroelectrics, and antiferroelectrics, the relationship between electric displacement vector and polarization intensity is shown as follows [39]:(4)$$D=P+\varepsilon_{0}E$$

To obtain the charge energy and discharge energy densities, the polarization axis and polarization-electric field (*P*-*E*) hysteresis loop should be integrated [40]. Formulas are described below:(5)$$U_{\mathrm{storage}}=\int_{0}^{P_{\max}}EdP$$
(6)$$U_{\mathrm{released}}=\int_{P_{r}}^{P_{\max}}EdP$$
where *P*_r_ is the residual polarization and *P*_max_ is the maximum polarization. From the above formulas, it can be noted that increasing the dielectric constant, polarization strength, and breakdown strength of the material promotes the dielectric to achieve higher energy storage density. In practical work, the material should have low dielectric loss and low conductivity to ensure application performance under an external electric field.

### 2.2. Energy Storage Efficiency

Energy storage efficiency is as important as energy storage density. Dielectrics are depolarized in the discharge process, resulting in the release of stored energy, which translates to energy loss (U_loss_) (Figure 2). Thus, energy storage efficiency is expressed as follows:(7)$$\eta =\frac{U_{\mathrm{released}}}{U_{\mathrm{released}}+U_{\mathrm{loss}}}=1-\frac{U_{\mathrm{loss}}}{U_{\mathrm{released}}+U_{\mathrm{loss}}}$$

If dielectrics have a high charge-discharge efficiency (*η*), they can achieve continuous discharge of high power density pulses. However, if *η* is low and U_loss_ is too high, the generated energy will increase the temperature of the capacitor, which leads to the internal structure‘s destruction, deterioration of energy storage characteristics, and reduction in lifespan.

The energy storage characteristics of dielectrics are closely related to the charge-discharge frequency. Their life depends not only on insulation strength, but also on discharge frequency under a specific charging electric field. In brief, dielectric materials should have low dielectric loss and low residual polarization.

## 3. Linear Polymer Dielectrics

Linear polymers can be classified as polar or nonpolar. Poly(methyl methacrylate) PMMA, polyetherimide (PEI), polythiourea (PTU), and polysulfone (PSF) are polar polymers; polypropylene (PP) and polystyrene (PS) are nonpolar polymers. Various polymers, including polycarbonate (PC), poly(phenylene sulfide) (PPS), poly(ethylene 2,6-naphthalate) (PEN), and poly(ethylene terephthalate) (PET), have been successfully developed and applied in commercial capacitor dielectrics [41,42,43,44,45,46].

### 3.1. Linear Polymer Filling and Blending

The conventional method employs inorganic fillers, such as ceramic particles, to fill the polymer matrix, resulting in a two-phase composite material composed of inorganic particles and polymer to improve the energy storage characteristics of the polymer-based dielectric [47]. As a result, low breakdown strength and significant dielectric loss are observed [48,49]. On the one hand, compatibility between the inorganic filler and the polymer matrix is poor, and the bonding characteristics of the organic and inorganic two-phase interface have a significant effect on the composites’ service life. On the other hand, a higher dielectric constant requires a higher filling content, which reduces the polymer’s flexibility and results in suboptimal processing performance [50,51].

By incorporating appropriate organic polymers and organic polymer semiconductors as organic fillers into linear polymer matrixes, it is possible to increase the dielectric constant and adjust and optimize the material’s energy storage characteristics. In this section, the above linear polymers’ filling and blending methods are considered.

#### 3.1.1. Polar Polymer Filling and Blending

Polyimide (PI) is an imide-containing aromatic ring polymer. It is widely used in the insulation, flame retardant, sound-absorbing, and coating materials industries [52]. It has a dielectric constant of approximately 3–3.5, a dielectric loss of approximately 10^−3^, and a dielectric strength of up to 200 MV/m. Kapton PI has an energy storage (U_e_) of 0.44 J/cm^3^, and *η* is 76% at 220 MV/m. Moreover, PI has a high glass transition temperature, so it has high temperature energy storage performance [46,53,54,55].

For example, Xu et al. improved the dielectric properties of PI by using thermal imidization and solution casting to fill the PI matrix with poly(arylene ether urea) (PEEU) as an organic substance. After blending, the maximum energy density was 5.14 J/cm^3^, *E*_b_~495.65 MV/m. As illustrated in Figure 3a, the blend film containing 15 wt% PEEU had a breakdown strength of 348.53 MV/m at 150 °C. Although the breakdown strength was 29.7% lower than at room temperature, it maintained its stability at high temperatures [56]. Li et al. also prepared PI/PSF composite dielectrics with suitable dielectric and mechanical properties via in situ polymerization and thermal imidization. At a frequency of 1 kHz, the dielectric constant was 6.40 and the dielectric loss was 0.0155. The 10.2% elongation at break was accompanied by a high *E*_b_ of 152.2 MV/m and a high U_m_ of 0.64 J/cm^3^ (Figure 3b) [57].

Polyetherimide (PEI) is a modified polymer of PI that is primarily defined by the imide ring on the main chain and the addition of ether bonds (-R-O-R-) to increase the molecule’s thermoplasticity [58,59]. It significantly improves the polyimide’s disadvantage as a poor thermoplastic. PEI also has good thermal stability (*T*_g_~217 °C) and low dielectric loss. At 150 °C and 200 MV/m, the discharge energy density can reach 0.5 J/cm^3^, and η is about 90% [52,60,61]. Meanwhile, PEI is widely used as an insulating material due to its superior electrical insulation properties and ease of processing.

Zhang et al., for example, mixed two high *T*_g_ polymers, PEI and poly(ether methyl ether urea) (PEMEU), in various mass ratios to create high-performance polymer dielectric materials with a high dielectric constant and low dielectric loss. When the ratio of PEI to PEMEU was 1:1, the blend polymer had a dielectric constant of 5.8 and a low dielectric loss (<1%). Meanwhile it demonstrated excellent dielectric properties over a wide temperature range (−70 °C to 150 °C) (Figure 3c,d). However, the breakdown and energy storage properties of PEI at high temperatures have not been studied [62].

Furthermore, adding molecular semiconductors into PEI to create all-organic composite materials has developed into a novel method. Li et al. reported mixing low concentrations of ITIC, PCBM, and DPDI into PEI. The three kinds of molecular semiconductors are named tetrakis(4hexylphenyl)-dihydrodithieno-s-indaceno dithiophene-2,8-diyl bis-methylidynebis [propanedinitrile]), phenyl C61 butyric acid methyl ester, and tetrakis(1-pentylhexyl)-bianthra diisoquinoline-octone.

The TSDC test demonstrated that the three organic polymer semiconductors could generate traps with high energy levels, which resulted in charge trapping (Figure 4a–c). Additionally, strong electrostatic attraction at high temperatures and fields could effectively fix the injection of electrons near the electrode/dielectric interface. Simultaneously, an electric field in the opposite direction of the applied field could be established to prevent charge injection and decrease the Schottky emission conduction current (Figure 4d–f). Finally, the maximum discharge energy density of PEI/PCBM was 4.5 J/cm^3^ and 3 J/cm^3^ at 150 and 200 °C, respectively; *η* was 90% (Figure 4g,h), which was superior to the current high-temperature energy storage dielectric (Figure 4i) [61].

**Figure 3 polymers-14-01160-f003:**
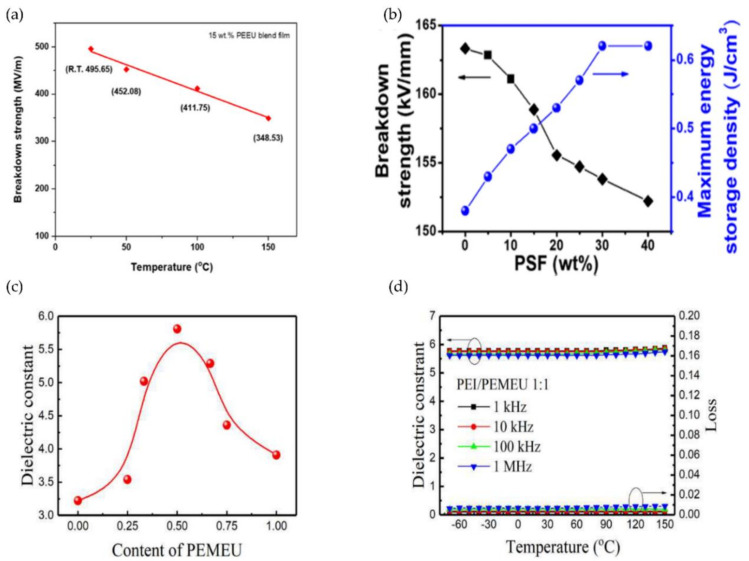
(**a**) Point line diagram of breakdown strength with temperature; (**b**) Breakdown strength and energy storage characteristics of PI/PSF films; reprinted with permission from [56,57]. Copyright (2021), Wiley. (**c**) the relationship between dielectric constant and blend composition (weight ratio of PEI: PEMEU) at 1 kHz; (**d**) The dielectric properties of a 1:1 mixture of PEI and PEMEU vary with temperature at different frequencies. Reprinted with permission from [62]. Copyright (2019), Elsevier.

Poly(methylmethacrylate) (PMMA) is a linear amorphous polymer with ε~3.49 and tanδ > 4% [63]. The glass transition temperature is between 100 and 130 °C [64,65]. PMMA has an energy storage density of only 1.25 J/cm^3^, and an efficiency of approximately 80% at 250 MV/m [66].

Poly(phenylene sulfide) (PPS) is a thermoplastic crystalline resin with excellent stability at high temperatures and resistance to corrosion. As a result, it is a widely used polymer material. However, its dielectric loss is less than 0.001 at 1 kHz at 100 °C. As a result, modifications to the molecular chain are required to make it stable at high temperatures [67,68,69].

Wu et al. recently synthesized polyoxafluoronorbornene (POFNB) via ring-opening metathesis polymerization with a Grubbs generation 2 catalyst. At 150 °C and 600 MV/m, the dielectric constant of POFNB reached 2.5 J/cm^3^ and the loss was less than 0.005, resulting in a discharged energy of 5 J/cm^3^ and an efficiency greater than 80%. Additionally, POFNB had a temperature coefficient of 0.016% °C^−1^, indicating a capacitive system with stable performance. Additionally, the author used density functional theory (DFT) to calculate the electronic densities of PEI, PP, PI, PEEK, and POFNB. The results indicated that the band gaps of these polymers were in the following order: PP > POFNB > PEEK > PEI > PI. Due to large bandgap of 5 eV and the flexibility of POFNB, the aromatic rings contributed to high π bonding energy levels.Finally, superior energy storage properties at high temperatures was shown [70].

PEI filling has a high energy storage efficiency (*η* > 80%), and a high energy storage density (U_e_ > 5 J/cm^3^) when used as a matrix for polar polymers. Additionally, although polymers such as PC and PPS exhibit low dielectric loss, high volume resistivity, high breakdown strength, and superior high-temperature stability, their low dielectric constant limits their access to high energy storage density.

**Figure 4 polymers-14-01160-f004:**
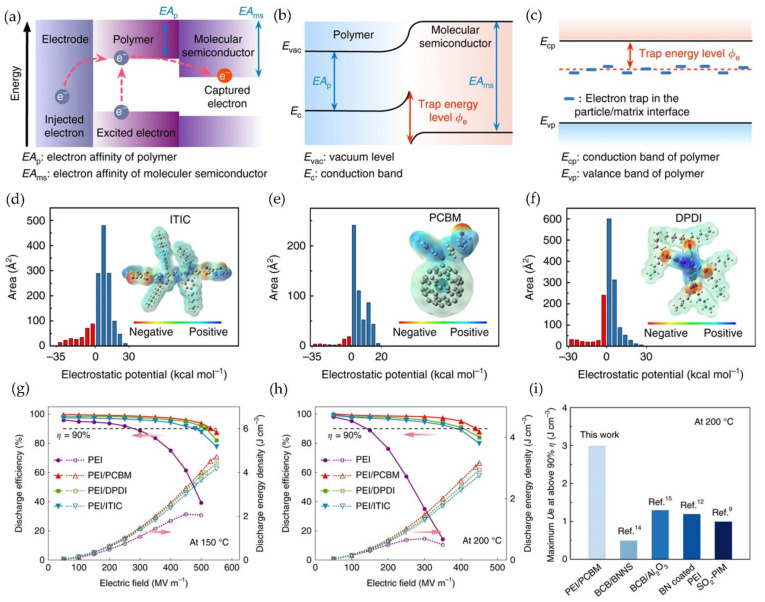
(**a**) Band diagram of charge transfer; (**b**) Schematic diagram of trap energy level introduced by molecular semiconductor; (**c**) Schematic diagram of trap energy levels of polymer/insulating particle nanocomposites; (**d**–**f**) Electrostatic potential distribution of three molecular semiconductors (ITIC, PCBM, DPDI); (**g**,**h**) Comparison of discharge energy density and charge-discharge efficiency of three blends and PEI at 150 °C and 200 °C; (**i**) Comparison of the maximum discharge energy density between PEI/PCBM and the most advanced high-temperature polymer medium at 200 °C when the efficiency is above 90%. Reprinted with permission from [61]. Copyright (2020), Springer.

#### 3.1.2. Nonpolar Polymer Filling and Blending

Polypropylene (PP) is a polymer composed of hydrocarbons. BOPP film, a biaxially oriented polypropylene film, is primarily used in commercial dielectrics. Additionally, it is a nonpolar polymer. At room temperature, the breakdown strength can reach 700 MV/m, the maximum operating temperature is 105 °C, and the dielectric constant and loss are approximately 2.25 and 0.01% at 1 kHz, respectively [71,72]. However, temperature has an effect on the discharge energy density and energy storage efficiency. When the temperature is increased from room temperature to 100 °C, for example, U_e_ decreases from 2.43 J/cm^3^ to 2 J/cm^3^, and *η* decreases from 99.3% to 80.5% [5,73].

In general, there are two methods for increasing the energy storage density of polypropylene-based polymers. Incorporating polar groups into the PP molecular chain is one strategy. Another strategy is to incorporate conductive fillers or nano-fillers into the PP-based nanocomposite during the filling process. By increasing the dielectric constant of PP, the energy density of the material is increased, which has critical implications for engineering applications. Yuan et al. increased the *ε*_r_ of PP by modifying it and synthesizing a copolymer with flexible hydroxyls (-OH 4.2mol% content) and polar groups (-NH_2_). The results indicated that this synthesis method was capable of increasing the dielectric constant of the modified PP to twice that of BOPP, maintaining a linear and reversible polarization curve, and achieving a final energy storage density of >7 J/cm^3^ at 600 MV/m [74].

Furthermore, Deng and Chen et al. grafted maleic anhydride onto polypropylene (MA-g-PP) to create hybrid dielectrics with pure PP films. The FTIR spectra of pure PP and PP/MA-g-PP demonstrated that polar groups were completely present in the mixed film, and that the grafting reaction proceeded successfully (Figure 5a). The dielectric constant increased with increasing MA-g-PP volume percentage, but the breakdown strength decreased. Due to the addition of the maleic anhydride group to the composite material to enhance the polarization of the dipole group, the film’s dielectric loss was relatively low. As shown in Figure 5b, 10 vol% MA-g-PP/PP processes energy at a maximum density of 1.96 J/cm^3^ with a maximum efficiency of 96% [75].

Li et al. created a binary PP composite by grafting PP with maleic anhydride. The breakdown strength of the DC circuit was 308.2 MV/m, the dielectric constant and losses were 2.56 and 0.4%, respectively, and the energy storage density was 1.076 J/cm^3^. As the electric field increased, space charges accumulated. Impurity ions accumulate near the electrode under high field conditions, resulting in ion transitions and ion polarization, thereby increasing dielectric loss. When impurity ions congregate close to the electrode in a high field, ion transition and polarization occur, resulting in increased dielectric loss. Due to the close relationship between the activation energy of ion jumps and the activation energy of molecular chain motion, optimizing the intermolecular interaction of PP is critical for increasing the energy storage density of composites [76].

PS has excellent water resistance and electrical properties at high frequencies. As a result, it is suitable for food packaging and electrical insulation. Additionally, it possesses excellent film-forming properties and can be used to fabricate films with a thickness less than 40 µm via biaxially oriented grafting [77].

As illustrated in Figure 5c, He and Zhang first prepared a low-compatibility poly[bis(4-cyanophenyl) 2-vinylterephthalate]/polystyrene (PBCN/PS) blend film. As a result, the rod-coil block copolymer polystyrene-b-poly[bis(4-cyanophenyl) 2-vinylterephthalate] (PS-b-PBCN) was mixed into it using a reversible addition-segment transfer polymerization method. Because PBCN has a high electron affinity, it inhibits electron migration over long distances while capturing a large amount of charge. While voids and defects are inevitable in the PBCN/PS blending system, the PS-b-PBCN/PS blending resembled a “core-shell” structure, which improved compatibility while increasing interface polarization. The results demonstrated that increasing PBCN content increased the dielectric constant and breakdown strength of the blended film. The energy storage density and efficiency were 2.16 J/cm^3^ and 90% at 295 MV/m, respectively, in the 30 wt% PS-b-PBCN/PS (Figure 5d) [78].

Although the energy storage density of composites can be increased through the filling and blending of nonpolar polymers PP and PS, the overall improvement is not significant. However, high energy storage efficiency can be maintained. Both types of all-organic polymers have the potential to significantly enhance dielectric and energy storage properties.

The filling and blending of linear polymers have been discussed previously; modification of polymer chains and processing of polymer dielectrics can still increase energy storage density. For example, Wei and colleagues synthesized a dipolar glass polymer, poly(2-(methylsulfonyl) ethyl methacrylate) (PMEMA), via free radical polymerization of methacrylate monomers. PEMSEA exhibited typical dipolar glass characteristics with a discharged energy density of 4.54 J/cm^3^ at 283 MV/m [79].

Despite their generally low dielectric constants, linear polymers continue to play a critical role in modern energy storage and power systems due to their outstanding advantages, such as high breakdown strength, low energy loss, long service life, and low cost.

The properties of linear polymer dielectrics are summarized in Table 1. Table 2 summarizes the chemical structures of linear polymer dielectrics.

**Figure 5 polymers-14-01160-f005:**
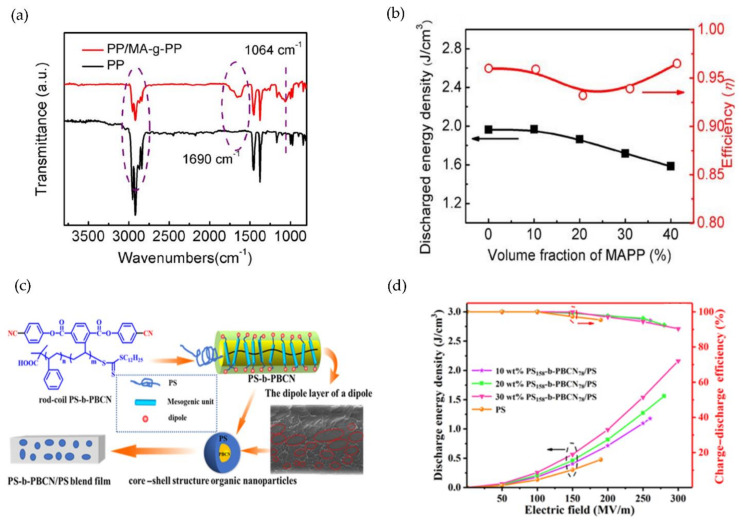
(**a**) Comparison in FTIR spectra between pure PP and 40 vol % MA-g-PP/ PP hybrid films; (**b**) The relationship between the discharge energy densities and efficiencies of PP/MA-g-PP hybrid films with various MA-g-PP volume fractions; reprinted with permission from [75]. Copyright (2017), Wiley. (**c**) Schematic illustration of the mechanism of PS-b-PBCN organic fillers in the PS matrix; (**d**) Energy storage properties of PS158-b-PBCN78/PS blends. Reprinted with permission from [78]. Copyright (2021), American Chemical Society.

## 4. Nonlinear Polymer Dielectrics

Ferroelectric polymers are the most frequently used nonlinear polymers in practice. Although the ferroelectric polymer exhibits a high dielectric constant, its low breakdown strength and energy storage efficiency preclude its use in capacitor dielectrics [80]. This is primarily due to the interaction of large domains and highly coupled dipoles in ferroelectric polymers, which alters the direction of the dipoles and polarization hysteresis, resulting in high dielectric loss and a decrease in discharge energy density and charge-discharge efficiency.

### 4.1. Nonlinear Polymer Blending

Poly(vinylidene fluoride) (PVDF) is most frequently used in nonlinear polymers in binary and ternary forms. For instance, poly(vinylidenefluoride-trifluoroethylene-chlorotrifluoroethylene) (P(VDF-TrFE-CTFE)) is a ferroelectric material that relaxes due to its high dielectric constant (50), a high *β* phase, and a low *E*_b_. However, due to its low breakdown strength, it is not suitable for electrical insulation [81,82,83].

Blending methods for balancing or optimizing the properties of polymers have been proposed. Regulating the interaction between the molecular segments of the two-phase interface improves the dielectrics’ mechanical properties and contributes to their dielectric and breakdown properties. Direct mixing of ferroelectric polymers and their derivatives enables precise control of their internal composition, crystallization behavior, and interface problems [84,85].

The simplest method is to combine PVDF and binary copolymer poly(vinylidenefluoride-trifluoroethylene) (P(VDF-TrFE)). Meng et al. discovered that when PVDF/P(VDF-TrFE) was blended, adding a small amount of P(VDF-TrFE) could increase the *β*-phase content and preferred orientation of PVDF. Thus, when compared to PVDF and P(VDF-TrFE), the composite material‘s dielectric constant after blending was increased by 80% and 30%, respectively, as was the residual polarization strength (*P*_r_). Additionally, by suppressing the Curie transition, the operating temperature range was widened, resulting in high-temperature stability for dielectrics [86,87].

As shown in Figure 6a, Gong et al. prepared poly(vinylidene fluoride)/poly(vinylidenefluoride-trifluoroethylene-chlorofluoroethylene) PVDF/P(VDF-TrFE-CFE) blend composite membranes and adjusted the PVDF ratio to increase the *ε*_r_ and *E*_b_ of the membranes. By incorporating terpolymer into PVDF, the two polymers became more compatible, thereby increasing the recombination and interfacial polarization effects in dielectrics. Finally, at 480 MV/m, the discharge energy density could reach 13.63 J/cm^3^ (Figure 6b) [88]. Zhang and Shen also used solution casting to create a film from PVDF and P(VDF-TrFE-CFE). When the blending system ratio was 60/40 vol%, U_e_ reached 19.6 J/cm^3^. The interface between the two polymers induced interface polarization by suppressing premature saturation of polarization. As a result, blended polymers outperformed pure PVDF and its terpolymer in terms of energy density and breakdown strength (Figure 6c–e) [89].

Chen et al. developed a hybrid relaxor ferroelectric P(VDF-TrFE-CTFE) terpolymer with positive ferroelectric properties P(VDF-TrFE). They demonstrated that a few copolymers could enhance relaxation polarization and that the resulting interface coupling effect could convert copolymer into a relaxant. On the other hand, as the copolymer concentration increased (15 wt%–20 wt%), the interfacial polarization decreased, the copolymer converted to normal ferroelectrics, and the reduction of *ε*_r_ resulted in polarization weakening. As a result, the ferroelectric response mechanism of the blends could be tuned effectively by adjusting the copolymer content and eliminating defects [90].

Moreover, Malik et al. blended P(VDF-TrFE)/P(VDF-TrFE-CTFE) to investigate the composites’ polarization behavior, which was enhanced when compared to P(VDF-TrFE-CTFE). The antiferroelectric properties of the blends increased the energy density. In the α = 0.1 and α = 0.2 blend systems, the blend properties changed from paraelectric to antiferroelectric behavior, and their storage-release energy density was also increased [91].

While copolymers or blends based on PVDF have a high energy density when subjected to high electric fields, they also have a high field loss. Wu and Zhang proposed the addition of low-energy-loss additives to reduce the amount of energy lost by blends when exposed to high electric fields. When poly (ethylene chlorotrifluoroethylene) (ECTFE) was added to poly(vinylidene fluoride-chlorotrifluoroethylene) P(VDF-CTFE), the hysteresis loss decreased from 35% to 7% compared with P(VDF-CTFE). Thus, the crosslinked blend polymer can act as a polarization domain inhibitor. Controlling the degree of crosslinking in polymer blends at the nanoscale is a critical step toward increasing energy storage density and minimizing high field loss (Figure 7) [92,93].

Finally, in a nonlinear polymer blend system, the intermolecular interaction reduces the mobility of the polymer chains, resulting in a change in carrier transport behavior. Additionally, the enhanced interfacial polarization effect and the combined effect improve the polymer’s energy storage characteristics and maintain the polymer‘s inherent dielectric properties and flexibility. 

### 4.2. Nonlinear Polymer Filling

During the early stages of research and development, the most frequently used method was the filling of nonlinear polymers. The filling material can be in the form of 0-dimensional particles (0D), the most frequently used of which are oxide and perovskite structure compound particles. Electrospinning, hydrothermal, and template methods are used to fill one-dimensional fibers (1D) and common fibrous perovskite structures. Two-dimensional flake fillers (2D) are comprised of nanoflakes, such as BN and MoS_2_. Peeling, chemical vapor deposition (CVD), and molten salt are all methods of preparation.

However, the filler’s size and parameters, as well as the microstructure, such as the degree of dispersion of particles, fibers, and flakes, void defects, and cavity defects, will affect the composite material’s breakdown performance and mechanical properties, thereby limiting the entire manufacturing process [50,51]. As a result, industrialization and mass production have been limited to the nonlinear polymer filled with 0D filler. Currently, researchers are focusing their attention on organic conductive filler/polymer composite dielectrics.

Organic conductive fillers have two advantages: a greater adjustment range for conductivity, and a stronger interaction between the filler and the matrix [24,94,95,96,97,98,99,100].

While it is difficult to prevent the formation of carrier tunneling channels in a composite material system with a high content of conductive particles, it is relatively easy to form conductive channels under low-content filling conditions. The organic conductive filler polyaniline (PANI) has a low elastic modulus and adjustable conductivity. It is also compatible with the matrix and maintains good dispersion even at high filling concentrations. For instance, Yuan et al. prepared PANI/PVDF composite dielectrics via a flowing method that demonstrated a steady increase in conductivity as the volumetric fraction of PANI (*f*_PANI_) increased (Figure 8a). When the doping amount of PANI approached the percolation threshold (*f*_c_ = 0.042), the dielectric constant increased rapidly, resulting in the formation of numerous micro-capacitive structures in the material (Figure 8b and Figure 9). These structures enabled the composite material to maintain a high energy density (7.2 J/cm^3^) within the PANI filling volume range [101,102].

By using PVDF as the matrix, an all-organic composite material is created, retaining the composite material‘s high breakdown strength, flexibility, and easy of processing.

Du and Dong, similarly filled poly(anthraquinone sulphide) (PAQS) particles with a wide band gap (3 eV) into PVDF to enhance the *E*_b_ and electric displacement. The overall preparation procedure was straightforward, requiring no chemical modification of the surface. As a result, composite materials‘ mechanical properties, such as elongation at break, mechanical strength, and toughness, were improved [103].

By incorporating elastomer/thermoplastic materials into nonlinear polymers, density and cost can be reduced, which benefits composites in practical applications. For example, solution cast composite dielectrics of methacrylate-butadiene-styrene/poly(vinylidene fluoride) (MBS/PVDF) were prepared. MBS rubber nanoparticles exhibited excellent flexibility, and their entanglement morphology had the potential to close gaps in the PVDF matrix, inhibit void defects, and increase breakdown strength. The dielectric constant ε_r_ decreased slightly as the MBS content increased, but *E*_b_ increased to 535 MV/m and the energy density reached 9.85 J/cm^3^ [104]. 

At the same time, Zheng et al. filled thermoplastic polyurethane (TPU) with varying degrees of hardness into PVDF to further study the elastomer, demonstrating that the lower the hardness of TPU, the more effective it was at improving breakdown strength. The conformation of the TPU soft chain interacted with the molecules in PVDF‘s amorphous region, effectively reducing defects and impeding matrix breakdown damage. According to the calculations, the maximum energy storage density was 10.36 J/cm^3^ [105].

Recently, a study used the core-shell structure to fill nonlinear polymers. For instance, MBS was mixed into the polymer matrix poly(vinylidene fluoride-hexafluoropropylene) (P(VDF-HFP)) as a filler. By incorporating rubber particles into MBS, the breakdown strength was increased and the electrical and mechanical properties were improved. However, an excessive amount of content will have a detrimental effect on the energy storage characteristics of MBS as its load capacity cannot withstand high electric fields. This work also demonstrated the effect of matrix-filler compatibility on the performance of all-organic polymer dielectrics [106].

Filling materials with nanofibers (1D) rather than nanoparticles (0D) improves their dielectric and energy storage properties. When organic fillers are not spherical but fibrous or tubular, the percolation threshold decreases. One-dimensional conductive fillers with high aspect ratios can significantly reduce the percolation threshold in composite material systems [51].

Wang et al. filled P(VDF-TrFE) with proton acid-doped polyaniline nanofibers. The percolation threshold of the composite medium was as low as 2.9 wt% for the fully doped polyaniline nanofibers. However, the degree of doping had a significant effect on the composite system’s percolation threshold, and the dielectric constant increased in its vicinity because of the formation of an interface between the organic conductive fiber and the polymer matrix [107].

Zhang and Cheng reported that when polypyrrole (PPy) was used to fill P(VDF-TrFE), an all-organic composite material with a high dielectric constant (2000) and a low percolation threshold (<8 wt%) was created. Additionally, when polarization occurred, a new dielectric relaxation process occurred, and the relaxation time decreased as the PPy content increased [96].

DC conductivity was found to be the most important factor in determining the percolation threshold, and neither the dielectric constant nor the AC conductivity should be used. The new relaxation process accounted for much of the dielectric loss. The loss observed during this process increased as the PPy contentration increased. Moreover, the PPy nanoclips were wettable with the polymer matrix. The glass transition process of the composites followed a single trend and decreased as the PPy content increased [108].

The matrix and filler can change each other, as in the case of P(VDF-TrFE-CFE) and P(VDF-TrFE-CTFE) filling.

Zhou et al. filled a polyurea (PUA) matrix with P(VDF-TrFE-CFE) to form all-organic composites. As a result, the energy storage density could be increased to 4.3 J/cm^3^, which was greater than the energy storage density of pure PUA (2.4 J/cm^3^), and the breakdown strength could be increased to 513 MV/m. The primary reason for the improvement was that the interaction of the two polymer’s molecular chains increased their dielectric constant. As a result, the energy storage density was increased while maintaining a high breakdown strength [109].

It was discovered that the dielectric constant of all organic composites was 90 times that of the matrix at 100 Hz in the study of PANI-P(VDF-TrFE-CTFE). The variation in the dielectric constant was consistent with seepage threshold theory. All-organic composites had a high dielectric constant that was flexible and variable. They had a similar elastic modulus when combined with an insulating polymer matrix and could signficantly reduce the high electromechanical response when subjected to a small electric field [98].

In addition, the introduction of aromatic polymers, such as polyurea (ArPU) and polythiourea (ArPTU), into the PVDF matrix has become a method for the preparation of all-organic composites.

The aromatic ring in aromatic polyurea (ArPU) exhibits good thermal stability. At room temperature, energy storage densities of 12 J/cm^3^ were obtained, while at 180 °C, densities of 6 J/cm^3^ were obtained [110,111,112,113]. Energy storage densities of 12 J/cm^3^ have been obtained at room temperature and densities of 6 J/cm^3^ have been obtained at 180 °C [110,111]. Meta-aromatic polyurea (meta-ArPU) was synthesized by varying the dipole moment and density of the two aromatic rings, increasing the dielectric constant, and decreasing the ferroelectric loss. A final energy storage density (13 J/cm^3^) with high efficiency (90%) was obtained [112]. Reducing the conduction loss enabled an *E*_b_ of up to 22 J/cm^3^ to be achieved at 10 MV/cm (Figure 10) [113].

Zhu et al. synthesized all organic dielectrics using ArPTU prepared via the polycondensation method and mixed with P(VDF-TrFE-CTFE)(15/85). ArPTU in polyvinylidene fluoride is an effective method. Its application can help reduce conductance loss and polarization hysteresis. As a result, a high breakdown strength (700 MV/m) and energy storage density (19.2 J/cm^3^) were achieved in the blended P(VDF-TrFE-CTFE)/ArPTU film, along with an efficiency of more than 85% [114].

While some progress has been made in the preparation (blending or filling) of nonlinear polymers, the crystalline morphology of monomers still requires improvement through research and development.

### 4.3. Multilayer-Structured Nonlinear Polymer

The basic principle behind the addition of spheroid (0D), fiber (1D), and platelet (2D) fillers to the polymer matrix to form high-k materials is to increase the interface area. Increases in the dielectric constant result in a decrease in the breakdown strength, thereby sacrificing the dielectric flexibility characteristics of polymer-based dielectrics [115,116]. The nonlinear polymers mentioned previously have been blended and filled to form all-organic composites. This was done to maintain a balanced relationship between the composite material‘s insulation and flexibility.

Recent research has discovered that a multilayer structure can resolve the synergistic relationship between the breakdown strength and dielectric properties of composite materials. The layered structure improves the dielectric energy loss in an electric field due to the advantages of layer-by-layer stacking and the cooperation of the layering effect. Additionally, it gradually decreases the migration rate of impurity ions in different layers, allowing for more combinations of different polymers.

As shown in Figure 11a, Jiang et al. proposed an electrospinning method for preparing multilayered P(VDF-HFP)/P(VDF-TrFE-CFE)(Co/Ter polymer) nanocomposites with a topological structure and phase composition (4, 8, and 16 layers). Local electric fields in multilayer structures were significantly weakened between the Co/Ter polymer layers, as demonstrated by phase-field simulations. As a result, the leakage current density was reduced, resulting in decreased conduction loss (Figure 11b). Significantly, at 600 MV/m, a high energy density of 20 J/cm^3^ was achieved (Figure 11c) [117].

Furthermore, the coaxial spinning method was used to prepare the core/shell composite structure of P(VDF-TrFE)/P(VDF-TrFE-CTFE). The C-F bonds were tightly coupled in this structure to increase the density of nanofibers, which exhibited enhanced mechanical, dielectric, and piezoelectric properties. At 12 Hz, both uniaxial and coaxial spinning had the highest dielectric constant, owing to the polarization and space charge effects at the internal Maxwell–Wagner–Sillars (MWS) interface. The two spinning structures had higher dielectric constants than pure polymers, which resulted from the conversion of ordinary ferroelectrics (FE) to relaxor ferroelectrics (RFE) via interface coupling, which reduced the crystal domains of the RFE [118].

Preparation of multilayer structural composites with PVDF and PVDF-based polymers is a standard procedure. Wang et al. prepared sandwich-structure composite dielectrics using a layer-by-layer casting method in which the top and bottom layers were PVDF and the middle layer was terpolymer P(VDF-TrFE-CTFE). The results indicated that as the terpolymer content increased, the dielectric constant and loss increased, and the maximum discharge energy density reached 20.86 J/cm^3^ [119].

Wei et al. designed sandwich structure dielectrics using PVDF/P(VDF-TrFE) blend dielectrics, with PVDF forming the outer layers on both sides. According to a comparative study of the effect of a PVDF and P(VDF-TrFE) blend structure and a sandwich structure on the composite dielectric properties, the optimal volume fraction of the ferroelectric phase in the blend film was between 10% and 30%, and the sandwich structure contained 40% P(VDF-TrFE). Finally a high U_e_ (24 J/cm^3^) and charge–discharge efficiency (>65%) were achieved. The sandwich structure ensured that the dielectric constants of each layer were distributed uniformly. The nonlinear polymer blend system modified the layer/interlayer interface by adjusting the volume fraction of the ferroelectric phase and balancing the breakdown strength and potential shift [120].

Among the three methods for preparing nonlinear polymer dielectrics, multilayer structure allows for the highest energy density (U_e_ > 20 J/cm^3^) and breakdown strength (*E*_b_ > 500 MV/m) in all-organic polymer dielectrics, but the energy storage efficiency is low. All-organic nonlinear polymers with filled structures have a high capacity for energy storage. For example, by filling the composites with aromatic polymers, the energy storage efficiency of the composites can exceed 90%, and ArPTU films can achieve ultra-high breakdown strengths (>1.1 GV/m) while maintaining an energy storage efficiency of 92.5% [114]. 

The energy storage density of blended polymers is typically equal to the sum of their two components‘ properties. While PVDF-based blends exhibit high energy storage density when subject to high electric fields, they suffer from high loss, which makes them unsuitable for engineering applications. Polarized microdomains created by a strong electric field result in significant losses. Loss results in a temperature increase, which has a detrimental efffect on the capacitor’s performance.

All-organic composite structures enable the rapid development of novel polymer-based energy storage materials. Other composite structures, such as ceramic/polymer sandwich structures, improve the dielectric properties but significantly reduce breakdown strength. Due to the significant dielectric difference between the two dielectrics, the system’s energy storage performances cannot be improved.

In order to provide readers with a more intuitive comparison, the key physical parameters of nonlinear polymer dielectrics including dielectric properties and energy storage properties are summarized in Table 3. The chemical structures of nonlinear polymer dielectrics are summarized in Table 4.

## 5. Linear/Nonlinear Polymer Dielectrics

As mentioned previously, ferroelectric polymers are the most frequently used nonlinear polymers in practice. They have a high dielectric constant, but their low breakdown strength and energy storage efficiency preclude their use as capacitor dielectrics [83,85,115]. Due to the interaction of large electric domains and high coupling dipoles in ferroelectric polymers, not only the dipole direction and polarization hysteresis are changed, but also the dielectric loss and discharge energy density are increased. 

When it comes to developing high-performance capacitor dielectrics, they are no longer limited to a single polymer material system but can be composed of a mixture of linear and nonlinear polymers of varying types.

### 5.1. Linear/Nonlinear Polymer Blending

Combining linear and nonlinear polymers effectively increases the breakdown strength and discharge energy density. PVDF, P(VDF-CTFE), PMMA, P(VDF-TrFE-CFE), and P(VDF-HFP) composite blends, for example, exhibit excellent energy storage performances.

The combination of linear and nonlinear polymers can significantly improve the dielectric properties of composite materials, and the combination of two different types of polymers can also significantly improve the energy storage performances.

Numerous studies have been conducted on the blending mechanisms of PMMA, PVDF, and their derivatives. For instance, Meng et al. combined PMMA and PVDF to increase the dielectric constant of the film to 12 at 100 Hz, which was significantly higher than PMMA (~2.9) and comparable to PVDF. Furthermore, the addition of PMMA converted PVDF from a nonpolar α phase to a polarity *β* phase, and PVDF/PMMA (60/40 wt%) exhibited a relatively high energy density [121]. 

Kang and Park prepared dielectrics from PMMA/PVDF blends using spin-coating and melt-quenching techniques. Due to the low current leakage caused by unsaturated hysteresis, the residual polarization strength Pr and RMS roughness decreased proportionately as the PMMA content in the blend film increased (Figure 12a,b). When PMMA content was 10-20 wt%, the blended ferroelectric film could be used in low-voltage organic FEFET [122]. Simultaneously, the space charge behavior of the two polymers revealed that the blend system’s amorphous region was composed of amorphous PVDF and miscible PMMA/PVDF. As the PMMA content increased, more miscible domains converted to continuous domains, increasing the charge transport capacity [123]. 

Chi et al. prepared a PMMA/PVDF composite dielectric using a solution blending method. PMMA volume content and heat treatment temperature were investigated in terms of compatibility. There was no effect on the crystalline phase structure when the heat treatment temperature was increased from 60 °C to 200 °C. However, as the temperature of the heat treatment increased, the crystallinity increased. The corresponding crystallinity at 60 °C, 100 °C, 150 °C and 200 °C was 14.47%, 17.62%, 19.28% and 21.49%, respectively. PVDF transitioned from α phase to *β* phase crystal as PMMA volume increased. The corresponding crystallinity of pure PVDF, 25 vol% PMMA/PVDF, and 50 vol% PMMA/PVDF was 40.89%, 29.51%, and 19.28%, respectively. The crystallinity of the composite material decreased as the PMMA content was increased due to the PMMA‘s dilution effect and good compatibility with PVDF. PMMA molecules inhibited the movement of PVDF molecular chains, and forming conductive channels within the blend was difficult. As a result, as the PMMA content increased, the breakdown strength increased as well. When the optimal heat treatment temperature was 150 °C, 50 vol% PMMA/PVDF reached 20.1 J/cm^3^ at 570 MV/m (Figure 12c,d) [124].

**Figure 12 polymers-14-01160-f012:**
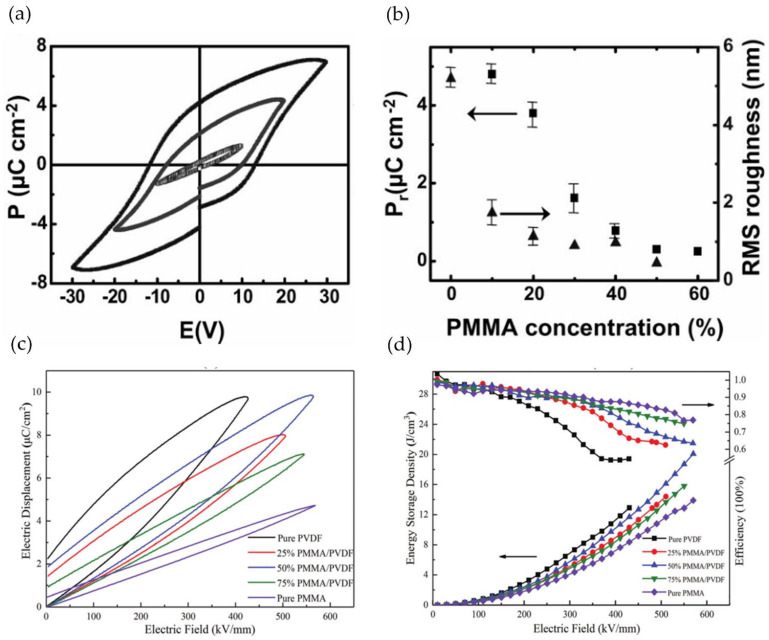
(**a**) The P-E loops of PMMA/PVDF (20:80) film ice-quenched and subsequently annealed at 150 °C; (**b**) Plots of P_r_ and RMS roughness as a function of the amount of the PMMA content. Reprinted with permission from [122]. Copyright (2009), Wiley. The electrical performances of PMMA/PVDF blended films with different content at 150 °C heat treatment (**c**) ferroelectric properties; (**d**) energy storage performances. Reprinted with permission from [124]. Copyright (2019), Royal Society of Chemistry.

The blending of PMMA and other PVDF copolymers performed admirably. Luo et al. used the Flory–Huggins model to investigate the Gibbs energy, miscibility and phase composition of binary mixtures. They dicovered that when PMMA reacts with P(VDF-HFP), a γ phase will form P(VDF-HFP). To maintain the phase’s stability and improve its energy storage characteristics, hydrogen bonds, van der Waals forces, and molecular chains on the interface were intertwined. Meanwhile, when combined with 42.6 vol% PMMA, a high U_e_ of 11.2 J/cm^3^ and a *η* of 85.8% at 475 MV/m were achieved [125].

Zhao et al. analyzed the dielectric relaxation characteristics of the PMMA/P(VDF-HFP) blend system. Near room temperature, the activation energy (*E*_a_), relaxation time (*τ*_HN_), and relaxation strength (Δε) of the composite material αc were significantly reduced. As the PMMA content increased, the degree of conductivity decreased more rapidly. As a result, the relaxation strength (ΔM), activation energy (*E*_a_), and relaxation time (*τ*_M_) of the dielectric modulus all increased. In blending, two α relaxation processes occurred that were not Debye relaxation processes [126].

The blending of PMMA and P(VDF-TrFE-CFE) was also due to the interaction of polymer molecular chains, which reduced the dielectric loss and the crystal grain size. The blending also improved the breakdown strength and energy storage density. When the PMMA content was 15 wt%, the composite exhibited a high discharge energy density of 9.3 J/cm^3^ at 520 MV/m, and the energy storage efficiency was 73% [127].

Research has focused on controlling the filler content, achieving a high dielectric constant at a low filler content, minimizing pores and defects, and optimizing the structure and properties of composites. For example, Dang et al. used PP as the matrix and thermoplastic PVDF as the filler to create PVDF/PP composites with a high dielectric constant via simple melt blending. Additionally, maleic anhydride-grafted polypropylene can be used as a compatibilizer to enhance the interaction of PP and PVDF, resulting in high-dielectric composite dielectrics [128].

Zhao et al. investigated series and parallel structures of PVDF and isotactic polypropylene (iPP). To increase their compatibility, 9.6 vol% and 18.3 vol% of polypropylene-graft-maleic anhydride (PP-g-MAH) were added as solubilizers. The dielectric temperature spectrum and dielectric spectrum revealed that filling iPP had little effect on the dielectric constant change and peak position of dielectric loss, indicating that iPP and PVDF were incompatible. Additionally, as the mixing time was increased, the dielectric constant of the composite material decreased, and the distribution of low-dielectric phases became more uniform. As a result, the parallel structure of the low-content iPP material was closer to that of the parallel structure. The parallel structure increased the breakdown strength, whereas the series structure made it easier to reduce the dielectric constant. As a result, the addition of the solubilizer maleic anhydride (PP-g-MAH) enhanced the dispersion and compatibility of material components and inhibited pore formation. Additionally, as the volume fraction of iPP and the time required for blending increased, the dielectric constant and breakdown strength of the composites improved [129].

In comparison to organic/inorganic composite dielectrics, all-organic composite structures offer lower preparation costs, softer machinery, easier processing, and a relatively simple process. PVDF was added to improve the dielectric constant via low-temperature chemical imidization of PI. PVDF particles dispersed uniformly can form space charges with PI, resulting in interfacial polarization [130]. 

### 5.2. Linear/Nonlinear Polymer Filling

Along with using high-dielectric fillers, conductive fillers can be used to enhance the energy storage characteristics of composite dielectrics. It has been reported that molecular semiconductors can be added to the above linear polymer filling to form all-organic composite dielectrics. Thus, linear/nonlinear polymers can still be used as matrixes and molecular semiconductors can be reintroduced as fillers.

Zhang et al. formed all-organic composites by adding organic polymer semiconductor [6,6]-phenyl C61 butyrate acid methyl ester (PCBM) as a filler to a PMMA/PVDF blend matrix. Because PCBM has a high affinity for electrons, it could effectively capture and excite carriers (the carrier concentration and migration speed were obtained by inhibition). This minimized the structural defects introduced by the molecules and improved the composite dielectric’s polarization and breakdown strength synergistically. By doping 0.9 wt% PCBM, excellent energy storage properties (U_e_~21.89 J/cm^3^; *η*~70.34%) were obtained under a 680 MV/m field intensity [131].

Electrospinning technology has become a new method for developing all-organic composite dielectrics by fusing linear and nonlinear polymers. As shown in Figure 13a, this work involves combining combinatorial-electrospinning and hot-pressing to fabricate all-organic dielectrics based on P(VDF-TrFE-CFE) with a ferroconcrete-like structure. First, hot-pressing the composite wet films converts them from a low T_g_ fluidized state to a continuous phase. Second, continuous polysulfone fibers (PSF_nfs), as a rigid scaffold, have excellent mechanical properties and are miscible with P(VDF-TrFE-CFE), thereby increasing the breakdown strength. 

According to the electromechanical breakdown mechanism, increasing the Young’s modulus results in improved mechanical properties and strengthens the breakdown force. Simultaneously, the high insulation characteristics of PSF and the large interface area carrier mobility, effectively suppress leakage current and conduction loss. The breakdown strength of P(VDF-TrFE-CFE)/PSF was shown to initially increase and then decrease with increasing PSF content. As a result, nanocomposites containing 30 vol% PSF achieved a high Weibull breakdown strength of 485 MV/m, more than 50% higher than the neat terpolymer (~320 MV/m) (Figure 13b,c) [132].

The combination of electrospinning and hot-pressing technology provides a new method for optimizing the energy density and breakdown strength of composite materials, allowing for the investigation of the relationship between the structure and properties of composites.

**Figure 13 polymers-14-01160-f013:**
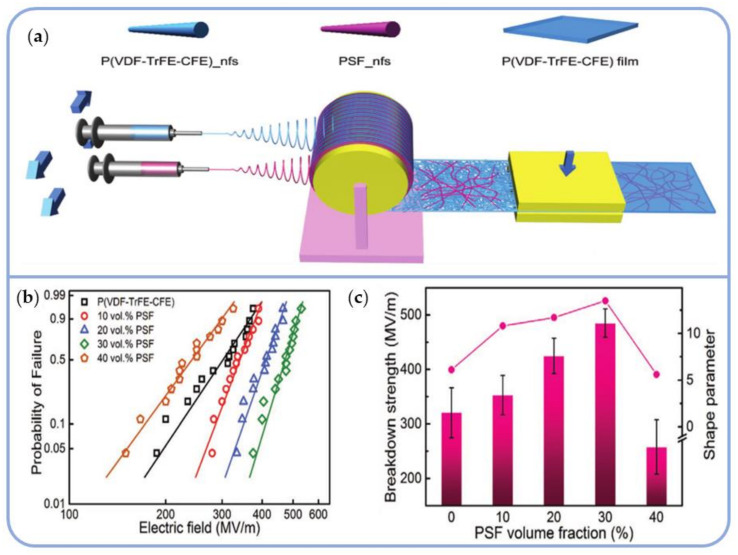
(**a**) Schematic diagram of the composite film prepared by combinatorial-electrospinning and hot-pressing; (**b**) The Weibull distribution for P(VDF-TrFE-CFE)/PSF with various PSF contents; (**c**) The Weibull breakdown strength and shape parameter β as functions of PSF volume fractions. Reprinted with permission from [132]. Copyright (2019), Wiley.

### 5.3. Multilayer-Structured Linear/Nonlinear Polymer 

Two difficulties arise when attempting to understand the mechanism of linear/nonlinear polymer blending and filling.

One is to elucidate the mechanism underlying the interface polarization effect. For example, as ferroelectric layers, PVDF-based polymers improve the local electric field. However, the primary polarization is still derived from Maxwelle–Wagnere–Sillars (MWS) polarization, which can be thought of as the dielectric constant or conductance interaction between adjacent dielectrics [133,134]. In the process, the interface polarization of the nonlinear multilayer dielectric can be established. As a result, obtaining nonlinear multilayer composite materials with a high discharge energy density via interface polarization is challenging.

The second issue is that the preparation of PVDF-based multilayer composite dielectrics results in interfacial corrosion during the preparation process. Because P(VDF-TrFE) and P(VDF-TrFE-CTFE) are highly compatible, dissolution of the polymer precursor (DMF) will corrode the PVDF composite layers. Thus, theoretical and practical difficulties arise during the preparation process.

To address the aforementioned issues, researchers evaluated the linear/nonlinear multilayer composite structure, which has advantages in terms of increasing the number of micro-interfaces, enhancing interface polarization, and increasing the breakdown strength. The double-layer structure is the simplest structure in the multilayer structure. Considering the relative positions of counter electrodes and interface relationships, some research has been undertaken on all-organic double-layer dielectrics.

Pei and Dang et al. evaluated double-layer structure dielectrics in PI, PP and PVDF experimentally. The study discovered that the electrodes‘ relative position had a direct effect on the breakdown strength of the composite dielectrics. When the k-layer was thinner and came into contact with the negative electrode, the breakdown strength increased. This was because when materials with different dielectric constants come into contact with the electrode, they redistribute the applied electric field and affect charge injection. On the other hand, a high k-layer can reduce the distortion of the electric field and increase the breakdown strength of double-layer dielectrics. The thickness change had a negligible effect on the electric field distortion [135].

Pei also investigated the electrode-dielectric interface’s effect on breakdown strength. To prepare an organic double-layer dielectric film, PMMA was used as the nano-interlayer and PVDF as the matrix. By varying the mass fraction of PMMA used in spin coating, the surface morphology of PVDF was modified. As the coating mass fraction increased, the film’s surface defects decreased and its flatness improved. The highest electric field strength and the lowest leakage current density were observed with a 1% PMMA coating film. Increased coating thickness resulted in a decrease in the rate of electric field distortion (Figure 14a), and an increase in the surface Young’s modulus. The phase-field method demonstrated that the PMMA coating inhibited the development of the breakdown phase and thus reduced the probability of breakdown. Covering defect holes has been shown to improve the insulation and capacitive performance of composite dielectrics (Figure 14b,c), alongside a significantly increased electric breakdown strength (767.05 MV/m) and energy density (19.08 J/cm^3^) [136]. 

In comparison to single-layer/double-layer dielectrics, the composite dielectric maintains good dielectric/energy storage characteristics due to the synergy between interfaces and the control of interlayer thickness in the multilayer structure. As a result, the multilayer structure’s design contributes to improved dielectric performance. The sandwich structure is a multilayer structure that is highly efficient, has low dissipation, has a long service life, and is extremely stable. When linear and nonlinear polymers are combined, it becomes possible to design all structured sandwich dielectrics.

Chen and Wang et al. developed sandwich polymer dielectrics composed of P(VDF-HFP)/PMMA/P(VDF-HFP) that combined the high insulation properties of linear dielectric PMMA with the properties of high polar ferroelectric polymer P(VDF-HFP) to achieve a balance of various performance parameters, such as dielectric loss and charge-discharge efficiency [137].

Additionally, they proposed PVDF as the outer layer on both sides and polyacrylate elastomers as the middle layer in order to customize the thickness of the middle layer and increase the energy storage density. When the thickness was 4 mm, the discharge energy density could reach 20.92 J/cm^3^ [138].

**Figure 14 polymers-14-01160-f014:**
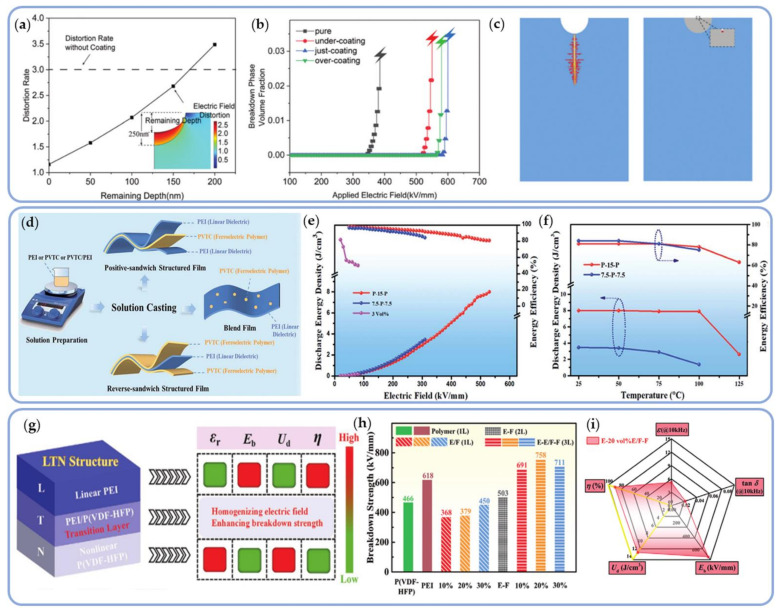
(**a**) Variations of electric field distortion and coating thickness; (**b**) Variations of breakdown phase volume fraction; (**c**) Development of a phase field method for simulating breakdown paths with and without coating. Reprinted with permission from [136]. Copyright (2021), Royal Society of Chemistry. (**d**) Schematic diagram of the preparation of positive sandwich structure, reverse sandwich structure and blended dielectrics; (**e**) The discharge energy density and efficiency of a positive sandwich structure P-15-P film, a reverse sandwich structure 7.5-P-7.5 film and 3 vol% PVTC/PEI blend film; (**f**) High-temperature energy storage characteristics of P-15-P and 7.5-P-7.5 at the highest electric field. Rreprinted with permission from [139]. Copyright (2021), Royal Society of Chemistry. (**g**) Schematic illustration of the dielectric energy-storage characteristics of the LTN structure; (**h**) Comparison of breakdown strength of PEI, P(VDF-HFP), single-layer PEI/P(VDF-HFP) blend composites, PEI-P(VDF-HFP) bilayer composite, and asymmetric trilayer composites; (**i**) The radar plots of the E-20 vol% E/F-F asymmetric trilayer composite. Reprinted with permission from [140]. Copyright (2021), Wiley.

As shown in Figure 14d, Wang et al. successfully prepared positive-sandwich structured dielectrics using simple layer-by-layer solution casting with PEI as the outer layer and poly(vinylidene fluoride-trifluoroethylene-chlorotrifluoroethylene) (PVTC) as the inner layer. The reverse-sandwich structured dielectrics were constructed with PVTC as the outer layer and the PEI as the inner layer. As an insulating layer, the linear PEI layer could effectively increase *E*_b_ and prevent charge injection into the electrode area. As the polarization layer, the intermediate layer (PVTC) preserved the composites’ polarization ability. Experimental results demonstrated that all-organic, sandwich-structured dielectrics could maintain a highest *E*_b_ of 530 MV/m and a maximum discharge energy density of 8.0 J/cm^3^ with a low level of 15 vol% PVTC between 25 °C and 100 °C, representing high-temperature stability (Figure 14e,f) [139]. 

Sun et al. developed asymmetric all-polymer trilayer composites after completing the symmetrical sandwich-structure composite dielectric. The linear and nonlinear layers were optimized, and a transition layer (T-layer) between them was introduced to create a new asymmetric (LTN) structure (Figure 14g). The L and N-layers were made of PEI and P(VDF-HFP), respectively, while the T-layer was made of a PEI/P(VDF-HFP) blend composite dielectric. By incorporating the T-layer, the dielectric constant could be increased, and the electric field distribution could be made more uniform. In comparison to single-layer (PEI, PVDF-HFP) and double-layer L-N-structured composite dielectrics (PEI-P(VDF-HFP)), LTN-structured dielectrics had the highest breakdown strength (758 MV/m). When the PEI content exceeded 30 vol%, *E*_b_ decreased due to PEI’s excellent insulation properties, which prevented the growth of breakdown paths (Figure 14h). Polarization at the interfaces between the layers increased the energy storage density even more. The addition of linear polymer PEI significantly decreased the conduction loss (*L*_c_) and ferroelectric loss (*L*_f_) of the composite material, resulting in a higher energy density of 12.15 J/cm^3^ and an efficiency of 89.9% (Figure 14i) [140].

Along with the sandwich structure described previously, a series of linear/nonlinear multilayer dielectrics have been developed using microlayer coextrusion technology. As illustrated in Figure 15a–c, Baer and colleagues used a multilayer coextrusion method to create PC/PVDF composite dielectrics with 2, 8, 32, and 256 layers. PC layers inhibited charge migration across the interface and minimized charge accumulation at the interface, thereby lowering the dielectric loss and increasing the energy storage density. Multilayer film was a viable method for effectively reducing impurity ion migration loss. The discharge energy density for 50PC/50PVDF (12 μm/380 nm) was 11 J/cm^3^ [141].

As illustrated in Figure 15c, this technology was used to create 65-layer PC/PMMA/P(VDF-HFP) dielectrics with a maximum breakdown strength of 880 kV/mm and a PMMA layer thickness of 25 nm. This produced a 25% increase in dielectric breakdown strength over the 33-layer PC/P(VDF-HFP). Futhermore, the 65-layer 46/8/46 PC/PMMA/P(VDF-HFP) exhibited significantly higher discharge energy densities, allowing for nearly 50% more energy storage than the 33-layer 50/50 PC/P(VDF-HFP) (Figure 15d) [142,143].

Maxwell–Wagner polarization (MWS) cannot account for the dielectric relaxation behavior caused by a change in the thickness of the P(VDF-HFP) layer. Nonetheless, the Sawada and Coelho models can shed light on the migration behavior of impurity ions in P(VDF-HFP) and explain why the dielectric properties of PC/P(VDF-HFP) layered dielectrics have improved [144].

It is necessary to control the effect of the thickness of different materials on the overall structure when fabricating multilayer polymer dielectrics.

Tseng prepared PSF/PVDF alternating multilayer dielectrics using a two-component multilayer coextrusion method, achieving a total thickness of 4.6 μm, 9.5 μm, and 11.6 μm for the 32-layer films, respectively. Similarly, the total thickness of the 256 film layers was 12 μm. The relationship between the multilayer structure and layer thickness revealed that the interface polarization generated by the space charge at the PSF/PVDF interface was stronger when both layers were thick (>100–200 nm). In contrast, the thinner both layers were (<100 nm), the weaker the interface polarization was found to be [145].

In conclusion, the thicker the PVDF is as a polarization layer, the more space charge can be polarized, and the thinner the PVDF is, the less migration loss of impurity ions can occur. The thinner the PSF, the stronger the electron conduction, thereby decreasing the total amount of interface polarization as an insulating layer. As the PSF becomes thicker, internal electron conduction is prevented.

**Figure 15 polymers-14-01160-f015:**
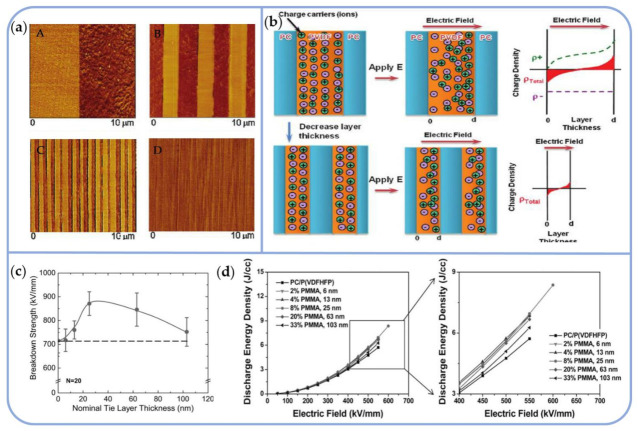
(**a**) The 2-A, 8-B, 32-C, and 256-D layer 50PC/50PVDF AFM phase images; (**b**) Schematic diagram of charge transfer. Reprinted with permission from [141]. Copyright (2012), American Chemical Society. (**c**) The breakdown strength of 33-layer PC/P(VDF-HFP), indicated as a dashed line, and the dielectric breakdown strength was evaluated for the 65-layer PC/PMMA/P(VDF-HFP) system as a function of PMMA layer thickness. (**d**) Discharge energy density comparison curves of the 65-layer PC/ PMMA/P(VDF-HFP) film and the 33-layer PC/P(VDF-HFP) film. Reprinted with permission from [143]. Copyright (2013), Wiley.

Futhermore, biaxially oriented BOPET/P(VDF-TrFE) multilayer structured capacitor dielectrics were fabricated using a forced assembly technique combining nanolayer coextrusion and biaxial orientation. This enabled regulation of dielectric loss in low fields and reduction in ion migration. These materials had energy densities of up to 16 J/cm^3^. Biaxially stretched dielectrics increase charge accumulation and thus the breakdown strength at the PET/P(VDF-TrFE) interface. Additionally, the edge P(VDF-TrFE) crystal causes a dielectric difference between PET and P(VDF-TrFE), thereby increasing the diameter and branching of the electrical tree [146].

Similar multilayer coextrusion techniques have been used to synthesize self-assembled, all-organic systems and other polymer organic-inorganic systems [147,148].

The multilayer structure primarily benefits from two advantages. The first is that the multilayer structure overcomes the breakdown, deterioration, and loss of flexibility that occurs when filler is added to a single matrix. Secondly, it can emphasize the physical characteristics of each layer, independently control the performance of each layers, integrate the advantages of each layer, and achieve synchronous improvement of the dielectric constant and breakdown strength. As a result, designing and fabricating multilayer structures for the purpose of obtaining polymer matrix composite dielectric materials with a high energy storage density has become a reliable method [149,150].

Table 5 summarizes the dielectric and energy storage properties of linear/nonlinear polymer dielectrics.

Among the three methods for preparing linear/nonlinear polymer dielectrics, the multilayer structure is unquestionably the most effective at increasing polarization and breakdown strength. Most preparation techniques involve hot-pressing and multilayer coextrusion. Energy storage efficiency of all-organic polymer dielectrics with sandwich structure is typically greater than 70%, and the maximum energy storage density increases to 20 J/cm^3^. When preparing the blend structure, the blending time and content of the blend will affect the material interface and internal defects, while simultaneously reducing dielectric loss and increasing the dielectric constant. In comparison to the other two structures, the filling structure had the highest energy storage density of 21.89 J/cm^3^ and the highest energy storage efficiency of 85%.

## 6. Conclusions

This literature review concentrated on the use of all-organic polymers in energy storage. The energy storage characteristics of polymer dielectrics were introduced. The polymer system was then classified as linear or nonlinear. Finally, the three structures (i.e., blending, filling, and multilayer) of all-organic polymer dielectrics were considered from the standpoint of preparation methods, and the following conclusions were drawn:Linear polymers are filled with organic polymers and molecular semiconductors, whereas nonlinear polymers are filled with organic conductive fillers or elastomer/thermoplastic materials. After filling, the polymer retains its excellent flexibility and processability. In addition, the excellent compatibility of the matrix and filler results in an increase in energy storage efficiency while decreasing ferroelectric loss;Nonlinear polymer blending utilizes the interface coupling effect to convert the initial nanophase into a relaxant or to induce the information of β phase, thereby increasing interfacial polarization and suppressing premature polarization saturation. The interaction of molecular chains during blending alters the crystallization behavior and modifies the crystal domain. This improves the hysteresis and energy loss associated with the large ferroelectric domains in the nonlinear polymer and the strong dipole coupling effect;The multilayer structure of all-organic polymers optimizes performance through a hierarchical combination. By varying the thickness and number of layers of the composites, the dielectric constant and electric field are redistributed. Additionally, by utilizing interface polarization and composite effects, the interaction between different interfaces can be enhanced and the synergy between balanced interfaces maintained.

To summarize, linear polymers have a low dielectric constant and loss, a high breakdown strength, and are highly efficient. Nonlinear polymers, on the other hand, have a high dielectric constant, ferroelectric loss, and a low discharge energy density, so they are inefficient. The collaborative design of two distinct types of polymers, the investigation of alternative preparation methods, and the synergy between matrix and filler all contribute to the development of new composite materials. Consequently, we need to develop a series of all-organic composite dielectrics with high energy storage density and insulation strength through material modification and structural design.

## 7. Future Suggestions

In the future, data processing techniques that combine first-principles simulation and machine learning will be used to screen out polymer matrices with high glass transition temperatures and high insulation. By combining breakdown simulation and high-throughput calculations, the optimal polymer composite structure form can be efficiently selected, and then controlled preparation of new composite materials can be accomplished by combining chemistry and materials methods. Additionally, the in situ characterization technology can be used to obtain information about the energy storage dielectric’s internal microstructure evolution, and combined with the temporal and spatial change law of the energy storage/dielectric performance under multiple fields, to investigate the structure-effect relationship.

Generally, we must integrate the benefits of multiple disciplines, optimize the microstructure of polymers, comprehend and predict the evolution of the structure in response to an applied field, and finally, improve the energy storage performance of polymer composites.

## Figures and Tables

**Figure 1 polymers-14-01160-f001:**
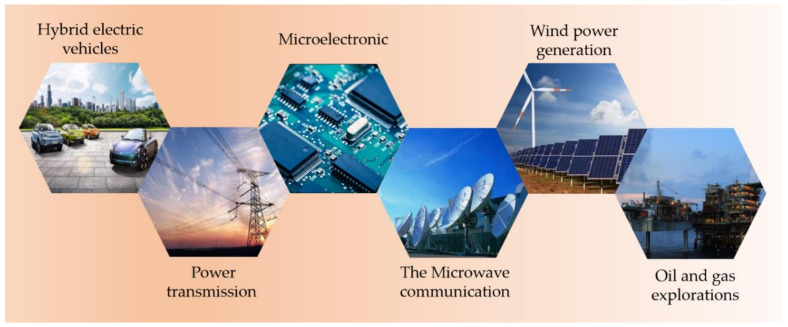
Application fields of polymer-based capacitors.

**Figure 2 polymers-14-01160-f002:**
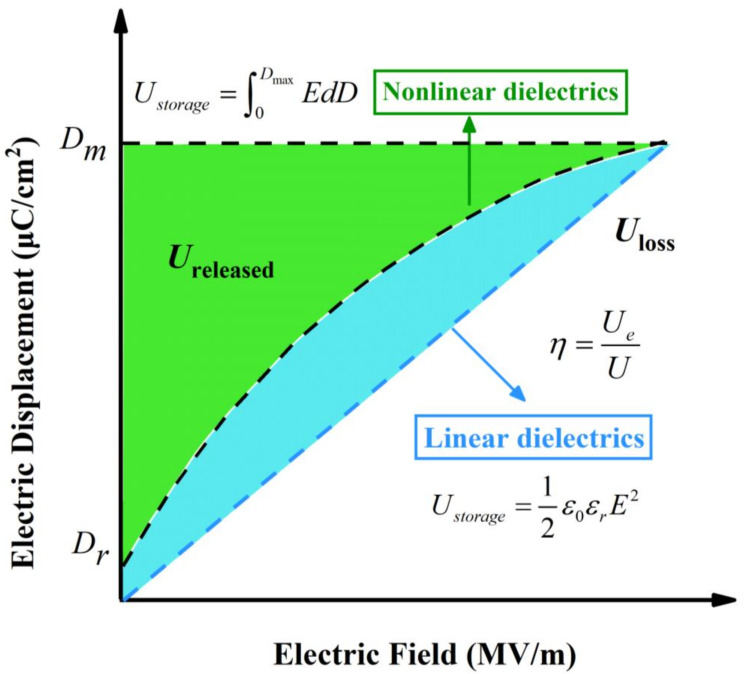
*D*-*E* loops of linear and nonlinear dielectrics. For linear dielectrics, energy density is determined by the triangular region (i.e., sum of both blue and green areas). For nonlinear dielectrics, energy density is determined by the green area, with the blue area representing energy loss. Energy storage efficiency is calculated from the green area to the triangular area.

**Figure 6 polymers-14-01160-f006:**
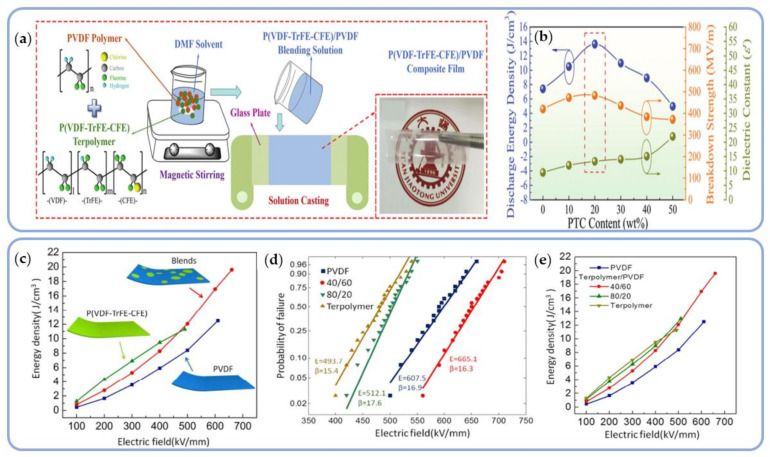
(**a**) Schematic diagram of PTC/PVDF preparation process and a photograph of the PVDF blended with 20 wt% PTC; (**b**) Contrast curve of discharge energy density, electric field intensity and dielectric constant; reprinted with permission from [88]. Copyright (2020), Royal Society of Chemistry. (**c**) Energy density comparison curve of blends, PVDF and P(VDF-TrFE); (**d**,**e**) Weibull distribution comparison curve and maximum discharge energy density curve of different content terpolymers and PVDF; reprinted with permission from [89]. Copyright (2016), American Chemical Society.

**Figure 7 polymers-14-01160-f007:**
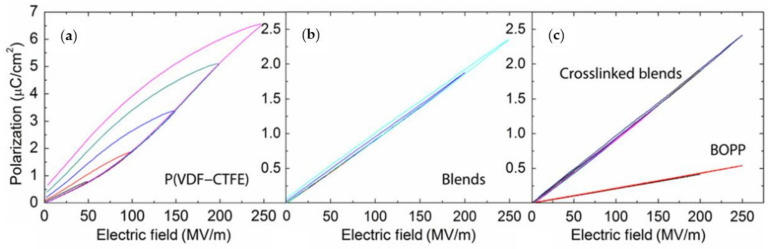
(**a**) Polarization-electric field (D-E) response of P(VDF-CTFE) 91/9mol%; (**b**) P(VDF-CTFE)/ECTFE 67/33 wt% blends; (**c**) P(VDF-CTFE)/ECTFE 67/33 wt% crosslinked blends with chemical crosslinking under unipolar-electric fields. Reprinted with permission from [93]. Copyright (2011), American Institute of Physics.

**Figure 8 polymers-14-01160-f008:**
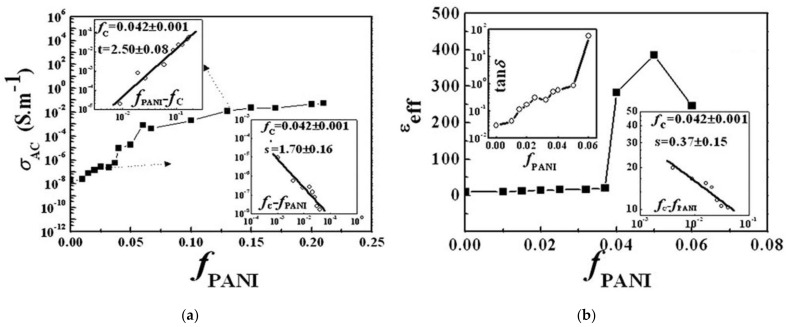
(**a**) The relationship between AC conductivity of PANI/PVDF composite and *f*_PANI_; (**b**) The relationship between the effective dielectric constant of PANI/PVDF composite and *f*_PANI_. Reprinted with permission from [101]. Copyright (2010), Royal Society of Chemistry.

**Figure 9 polymers-14-01160-f009:**
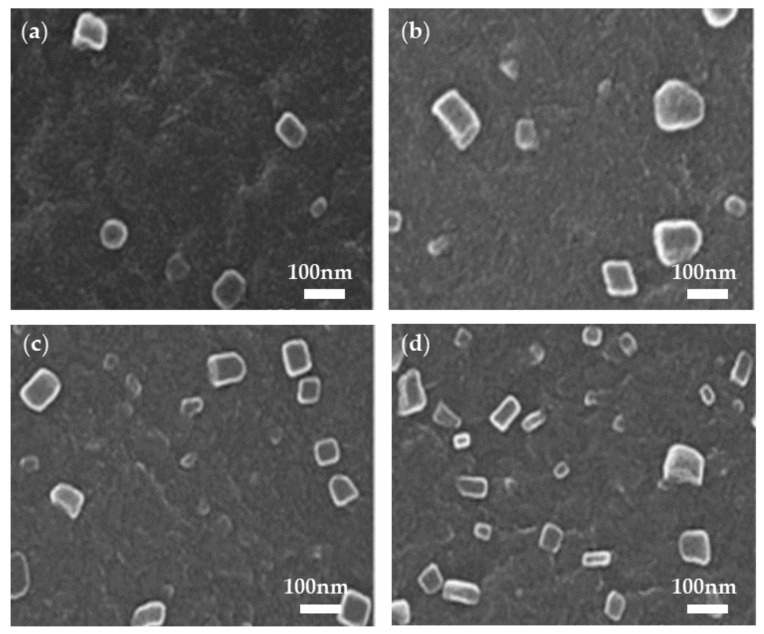
(**a**) SEM images of PVDF films with *f*_PANI_ = 0.015; (**b**) *f*_PANI_ = 0.040; (**c**) *f*_PANI_ = 0.050; and (**d**) *f*_PANI_ = 0.060. Reprinted with permission from [101]. Copyright (2010), Royal Society of Chemistry.

**Figure 10 polymers-14-01160-f010:**
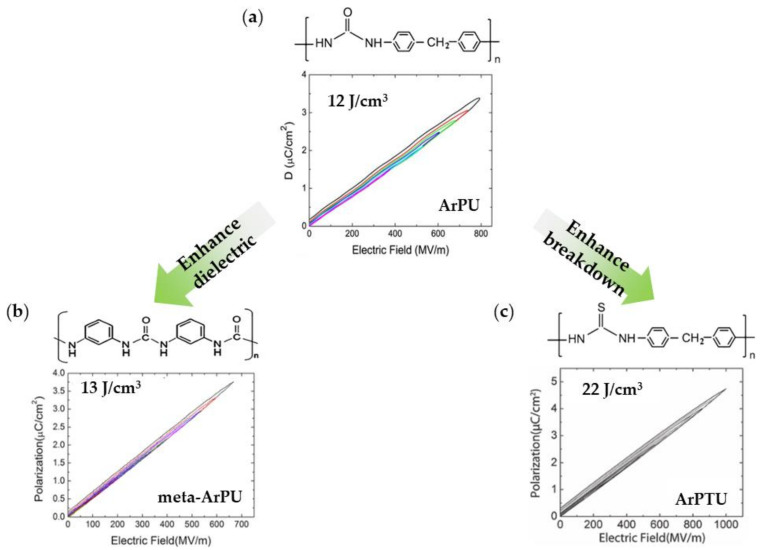
(**a**) The P-E loop with molecular formula ArPU, (**b**) meta-ArPU, and (**c**) ArPTU. Adapted from [110,111,112,113] with permission. Copyright (2017), Royal Society of Chemistry.

**Figure 11 polymers-14-01160-f011:**
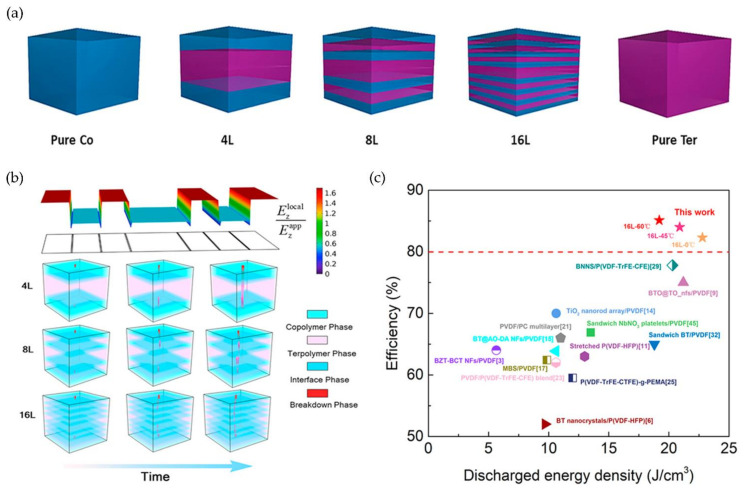
(**a**) Schematic diagram of multilayer structure, pure copolymer and pure terpolymer; (**b**) Comparison chart of discharge energy density and efficiency; (**c**) Schematic diagram of breakdown path evolution with time by phase-field simulation. Reprinted with permission from [117]. Copyright (2018), Elsevier.

**Table 1 polymers-14-01160-t001:** Dielectric and Energy Storage Properties of Linear Polymer Dielectrics.

Category	All-Organic Polymers	Dielectric Constant (at 1 kHz)	Dielectric Loss (at 1 kHz)	Breakdown Strength (MV/m)	Operating Temperature (°C)	Energy Density (J/cm^3^)	Efficiency (%)	Ref
Linear Polymer Dielectrics	PI	3–3.5	0.13	200	145–250	1.4	76	[54]
	PEI	3.3	0.3	400	150–200	0.5–1.5	90	[58,59,60,61]
	PMMA	3.49	0.04	250	RT	1.25	80	[63]
	PC	2.8	0.001	520	−50–200	0.5–1	95.8	[60]
	PPS	3	<0.001	220–550	−55–200	-	-	[68]
	PEN	3	0.004	300	RT	-	-	[60]
	PET	3.3	0.003	500	RT	1–1.5	-	[52]
	POFNB	2.5	<0.005	600	150	5	80	[70]
	PP	2.25	0.0001	620	−55–105	2.43	99	[71,72,73]
	PS	2.7	-	450	RT	5	-	[77]
	PMSEMA	10.5	0.02	283	RT	4.54	-	[79]
Polar Polymer Filling and Blending	PI/PEEU(15/75wt%)	4.78	0.00299	495.65	−50–250	5.14	-	[56]
	PI/PSF(40/60wt%)	6.4	0.0155	152.2	RT	0.64	-	[57]
	PEI/PEMEU(50/50wt%)	5.8	0.0087	-	RT	-	-	[62]
	PEI/PCBM	3.25	0.002	550	150	4.5	89	[61]
	PEI/DPDI	3.4	0.003	550	150	4.3	80	
	PEI/ITIC	3.3	0.002	550	150	4.1	78	
Nonpolar Polymer Filling and Blending	PP-OH(4.2mol%)-NH_2_	4.5	-	600	RT	>7	-	[74]
	PP/MA-g-PP(90/10vol%)	2.1	0.01	450	RT	1.96	96	[75]
	PP/PP-g-MAH(95/5vol%)	2.56	0.004	308.2	RT	1.076	-	[76]
	PBCN/PS(10/90wt%)	4.4	0.028	260	RT	1.68	85	[78]
	PS-b-PBCN/PS(30/70wt%)	5.8	0.018	295	RT	2.16	90	

Abbreviation: RT—room temperature.

**Table 2 polymers-14-01160-t002:** Molecular Structure of Linear Polymer Dielectrics.

Category	All-Organic Polymers	Compound Structure	Ref
Linear Polymer Dielectrics	PI	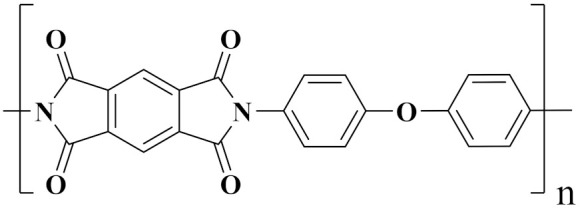	[52,54]
PEI	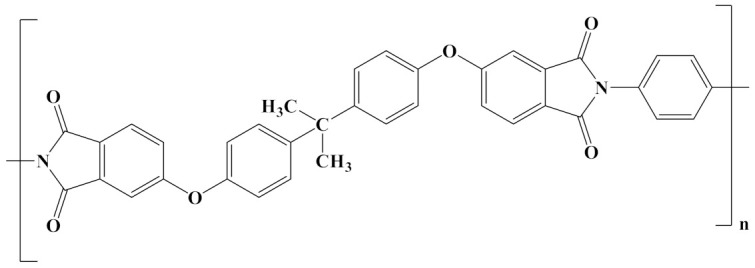	[52,58,59,60,61]
PMMA	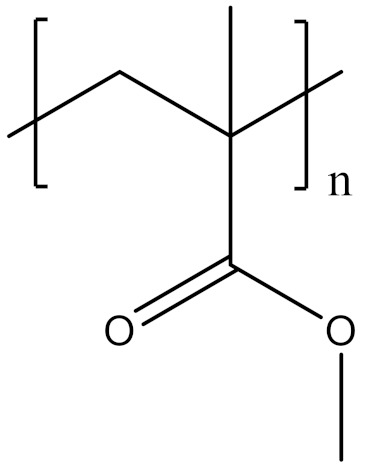	[63]
PC	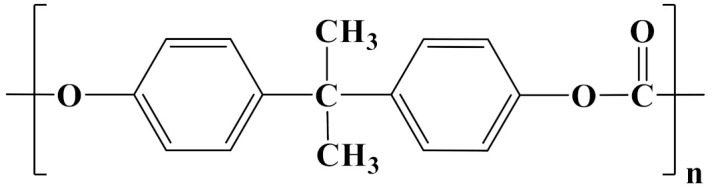	[52,54]
PPS	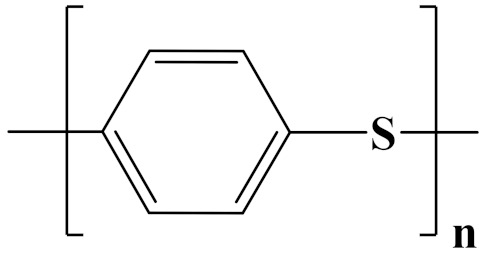	[52,58,59,60,61]
PEN	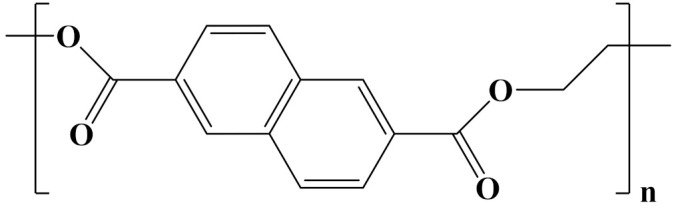	[52,63]
PET	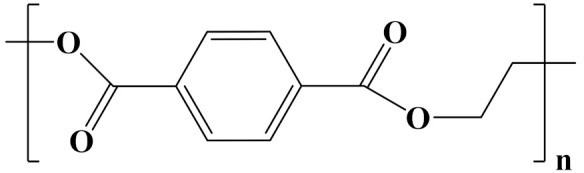	[52]
POFNB	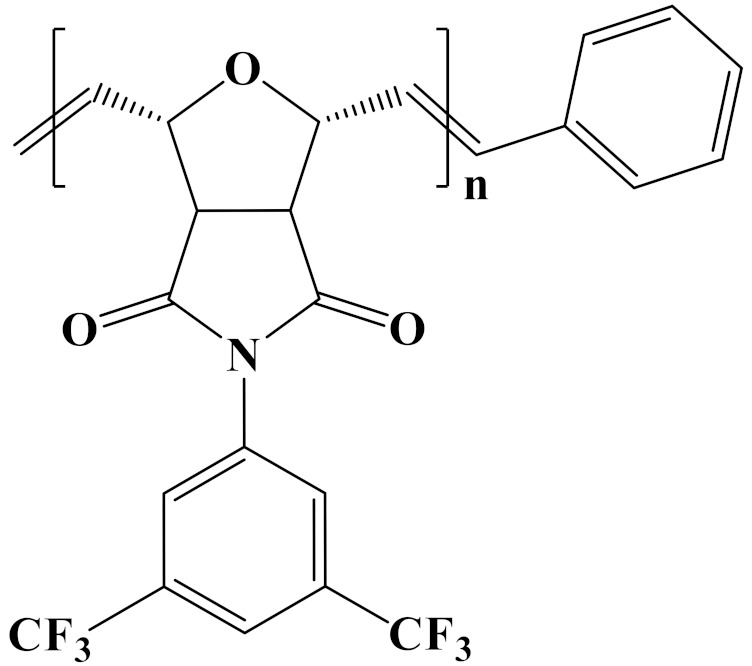	[70]
PP	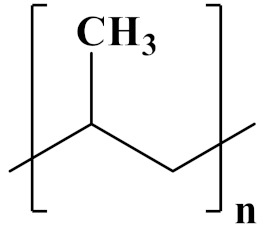	[52,71,72,73]
PS	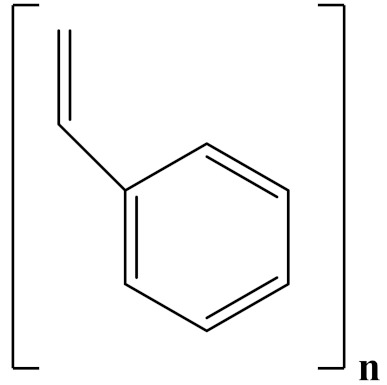	[77]
PMSEMA	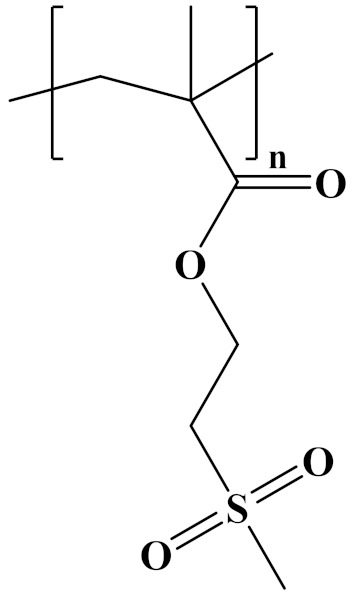	[79]
PEEU	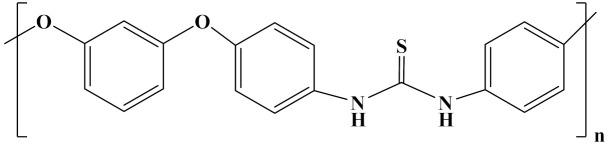	[56]
PSF	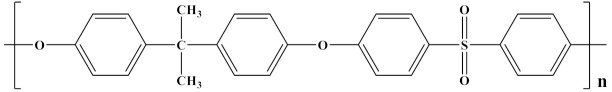	[57]
PEMEU	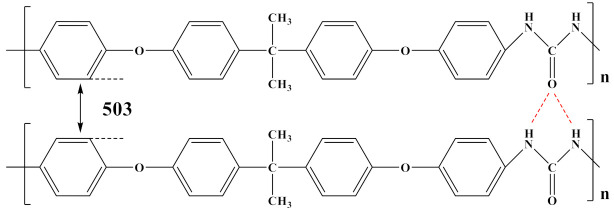	[62]

**Table 3 polymers-14-01160-t003:** Dielectric and Energy Storage Properties of Nonlinear Polymer Dielectrics.

Category	All-Organic Polymers	Dielectric Constant (at 1 kHz)	Dielectric Loss (at 1 kHz)	Breakdown Strength (MV/m)	Operating Temperature (°C)	Energy Density (J/cm^3^)	Efficiency (%)	Ref.
Nonlinear Polymer Dielectrics	PVDF	10	4	150–570	125	2.8	59	[81]
	P(VDF-HFP)	8.8	0.08	440	RT	10.4	65	[82,106]
	P(VDF-TrFE)	8.5	4.2	320	RT	1.3	47.71	[93]
	P(VDF-CTFE)	13	6	>600	RT	25	63	[83]
	P(VDF-TrFE-CFE)	50	4	350	RT	13	72	[83]
	P(VDF-TrFE-CTFE)	16	7.6	>500	RT	9.12	76.1	[81,82]
Nonlinear Polymer Blending	PVDF/P(VDF-TrFE) (70/30vol%)	23	0.04	200	RT	-	-	[86]
	PVDF/P(VDF-TrFE-CFE) (20/80wt%)	13	0.04	480	RT	13.63	-	[88]
	PVDF/P(VDF-TrFE-CTFE) (60/40vol%)	30	0.08	640	RT	19.6	-	[89]
	P(VDF-TrFE)/P(VDF-TrFE-CTFE) (60/40vol%)	38	0.1	400	RT	-	-	[90]
	P(VDF-TrFE)/P(VDF-TrFE-CTFE) (10/90vol%)	35	0.12	270	RT	6	-	[91]
	P(VDF-CTFE)/ECTFE (67/33wt%)	7	0.01	250	RT	2.8	-	[93]
Nonlinear Polymer Filling	PANI/PVDF (5/95vol%)	385	0.85	60	RT	6.1	-	[101]
	PANI/PVDF	-	-	-	20–150	-	-	[102]
	PAQS/PVDF (3/97wt%)	13	0.048	560	RT	18	70	[103]
	MBS/PVDF (12/88vol%)	7.8	0.02	535	RT	9.85	62.4	[104]
	TPU/PVDF (3/97vol%)	8.5	0.02	537.8	RT	10.36	-	[105]
	MBS/PVDF-HFP (8/92vol%)	6.75	0.06	515	RT	12.33	-	[106]
	PANI/P(VDF-TrFE) (2.9/97.1wt%)	50	0.1	25	RT	-	-	[107]
	PPy/P(VDF-TrFE) (8/92wt%)	2000	8	-	−60–140	-	-	[96]
	PANI-P(VDF-TrFE-CTFE)	>7000	0.7	16	RT	0.18	-	[98]
	ArPU/PVDF	4.2	0.005	800	25–200	>12	>90	[111]
	Meta-ArPU/PVDF	6	0.015	670	0–160	13	91	[112]
	ArPTU/PVDF	4.5	0.01	1000	0–150	22	92	[113]
	ArPTU/P(VDF-TrFE-CTFE) (15/85wt%)	11.3	0.01	700	RT	19.2	85	[114]
	PUA/P(VDF-TrFE-CFE) (70/30vol%)	5.3	0.05	513	RT	4.3	-	[109]
Multilayer-Structured Nonlinear Polymer	P(VDF-HFP)/P(VDF-TrFE-CFE) (22.9/77.1vol%)	10–53	-	600	0-60	20	85.1	[117]
	P(VDF-TrFE)/P(VDF-TrFE-CTFE) (core/shell)	3.1 *	0.015 *	-	RT	-	-	[118]
	PVDF/P(VDF-TrFE-CTFE)/PVDF (37.5/25/37.5vol%)	12 *	0.04 *	599	RT	20.86	65	[119]
	PVDF/(PVDF/P(VDF-TrFE)/PVDF (30/40/30vol%)	12	0.03	582	RT	23.4	>65	[120]

Abbreviation: RT—room temperature. * represents approximate values.

**Table 4 polymers-14-01160-t004:** Molecular Structure of Nonlinear Polymer Dielectrics.

Category	All-Organic Polymers	Compound Structure	Ref.
Nonlinear Polymer Dielectrics	PVDF	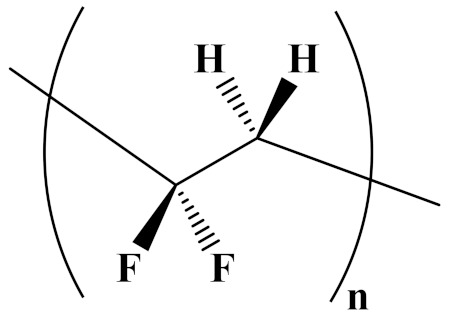	[81]
	P(VDF-HFP)	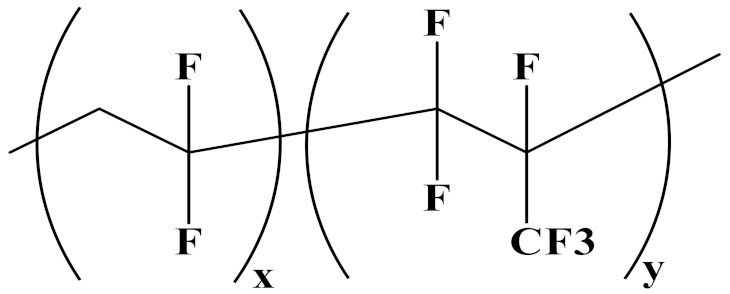	[82,106]
	P(VDF-TrFE)	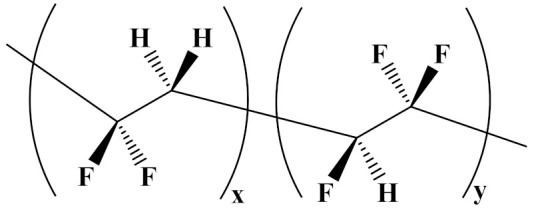	[93]
	P(VDF-CTFE)	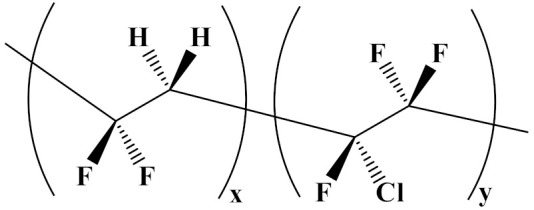	[83]
	P(VDF-TrFE-CFE)	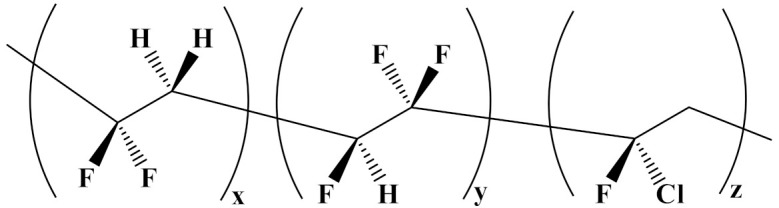	[83]
	P(VDF-TrFE-CTFE)	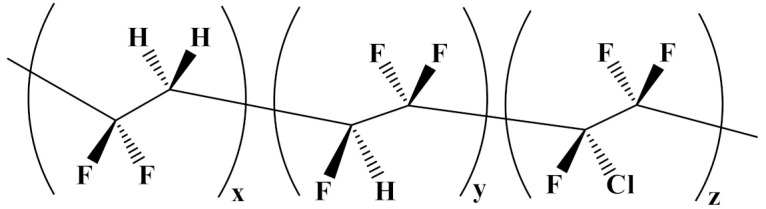	[81,82]

**Table 5 polymers-14-01160-t005:** Dielectric and Energy Storage Properties of Linear/Nonlinear Polymer Dielectrics.

Category	All-Organic Polymers	Dielectric Constant (at 1 kHz)	Dielectric Loss (at 1 kHz)	Breakdown Strength (MV/m)	Operating Temperature (°C)	Energy Density (J/cm^3^)	Efficiency (%)	Ref.
Linear/Nonlinear Polymer Blending	PVDF/PMMA (60/40wt%)	7.8	0.07	400	RT	9	-	[121]
	PMMA/PVDF (50/50vol%)	5.7	0.04	570	150	20.1	63.5	[124]
	P(VDF-HFP)/PMMA (57.4/42.6vol%)	6.5	0.017	476	RT	11.2	85.8	[125]
	PMMA/PVDF-HFP (30/70vol%)	-	0.05	542.6	−50–200	4.2	80.4	[126]
	PMMA/PVDF-TrFE-CFE (15/85wt%)	19	0.05	520	−40–100 *	9.3	73	[127]
	PP/PVDF (20/80wt%)	3.58	0.007	-	−50–150 *	-	-	[128]
	iPP/PVDF (39.7/60.3wt%)	7	0.02	325	−50–150	-	-	[129]
	PI/PVDF (50/50wt%)	7.745	<0.12	-	RT	-	-	[130]
Linear/Nonlinear Polymer Filling	PMMA/PVDF +0.9wt%PCBM	4.8	0.037	680	RT	21.89	70.34	[131]
	PSF_nfs/P(VDF-TrFE-CFE_nfs (30/70vol%)	30	0.05	485	RT	-	85	[132]
Multilayer-Structured Linear/Nonlinear Polymer	PP/PVDF	-	-	649.31	RT	-	-	[135]
	PI/PVDF	-	-	525	RT	-	-	[135]
	PMMA/PVDF (1/99wt%)	6.8	0.03	767.05	RT	19.08	60	[136]
	PVDF-HFP/PMMA/PVDF-HFP (35/30/35vol%)	7	0.065	-	RT	20.3	>80	[137]
	PVDF/PP/PVDF (37.5/25/37.5vol%)	9	0.06	438	RT	20.92	72	[138]
	PEI/PVTC/PEI (42.5/15/42.5vol%)	6	0.02	504	25–125	8	80	[139]
	PVTC/PEI/PVTC (17.5/65/17.5vol%)	22	0.06	345	25–125	5	75	
	PEI/(PEI/P(VDF-HFP))/P(VDF-HFP) (20/60/20vol%)	5	0.02	758	RT	12.15	89.9	[140]
	PC/PVDF (50/50vol%)	-	0.03	400	25–125	11	-	[141]
	PC/coPVDF	9	-	>600	RT	14	>60	[142]
	PC/PMMA/P(VDF-HFP) (46/8/46vol%)	4.98	0.018	850	-	8.36	-	[143]
	PC/PVDF-HFP (50/50vol%)	10	-	-	25–100	-	-	[144]
	PSF/PVDF (50/50vol%)	-	-	416	RT	-	-	[145]
	BOPET/PVDF-TrFE (50/50vol%)	7	-	450	120–140	16	-	[146]

Abbreviation: RT—room temperature. * represents approximate values.

## Data Availability

Not applicable.

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
