# Peer review of "Energy Storage Application of All-Organic Polymer Dielectrics: A Review"

_polymers, 2022, doi:10.3390/polym14061160_

Round 1

Reviewer 1 Report

A review «Energy Storage Application of All-Organic Polymer Dielectric» is devoted interesting and relevant topic. The application areas of polymer materials are expanding every year. One of the perspective modern fields for dielectric polymers use is in modern electronic and electrical systems. The introduction contains information about the existing shortcomings of polymers for the specified application, the focuses of researchers in the development of polymeric materials and ways to achieve the required properties. Section 2 describes in detail the design parameters, the formula used to calculate, illustrates the types of dielectrics depending on the conditions. The following are sections on specific types of linear and nonlinear dielectrics with different fillers and their characteristics. Special attention is paid to multilayer structures. The review contains a large number of figures, which makes it easier to understand the material. Section “Future Suggestions” draws attention. This is the perfect result of all the work. I would especially like to note the format for presenting data in the form of tables. The number of links used is enough. The conclusion briefly summarizes what has been written above.

Author Response

Dear Polymers Reviewers,

Re:Resubmission of Manuscript ID: polymers-1597243

Thanks for sending us your comments, which are extremely valuable and helpful for improving our manuscript titled “Energy Storage Application of All-Organic Polymer Dielectrics:A Review”. Thank you very much for giving us another chance to revise the paper. Your valuable suggestions and comments will be very helpful to our manuscript. I really appreciate that you took time out to review our manuscript. All the participating authors have discussed the comments carefully and reached a consensus. We have made the corresponding revisions in the manuscript and highlighted in red. In addition, we have adjusted the layout of the manuscript according to the requirements. After adopting your suggestions, we firmly believe that our manuscript will be greatly improved.

We have further taken this chance to carefully polish the manuscript and resubmit the revised manuscript to ‘Polymers’.

Thank you for your consideration.  

Sincerely yours,

Yu Feng

Reviewer 2 Report

In this manuscript, Feng and coauthors have reviewed the field of all-organic polymer dielectrics for energy storage in a concise manner. The authors first briefly introduce the current development of all-organic polymer dielectrics and three methodologies to construct the related polymeric composites, present some fundamental knowledge about energy density and energy storage efficiency and then categorise these polymeric composites into linear polymer-based one and nonlinear polymer-based one, in which the author sub-categorise the content using three different methodologies of construction. All-organic polymer dielectrics become a growing hot research topic, not only in the field of polymer chemistry but also in the fields of energy materials. The authors here generate this updated review, which makes this reviewer believe that it will bring considerable attention from readers.

Overall, the manuscript is written in a relatively proper way, and employs clear and coherent discussions. Herein, this reviewer suggest that it can be accepted by Polymers after addressing the issues listed below.

  • The authors were trying to summarise the current process of this field using certain parameters, including energy density and energy storage efficiency, which is good. However, it seems that they fail to emphasise these parameters when discussing the progress, especially when they state the comparison among various polymers. It would be great for the authors to stick to these parameters and discuss the difference and reason in detail.
  • As the authors discuss various types of polymers in the manuscript and use quite a few abbreviations for polymers, they should have given the full names of these polymers as they appear at the first time. Meanwhile, chemical structures of these polymer should also have presented.
  • The authors summarise some crucial properties of all-organic polymer dielectrics in Table 1, which is located at the end of the manuscript. It could be more attractive if the authors bring this table out earlier.
  • Correct some typos and polish the writing. It is very difficult for this reviewer to read through this manuscript.

Author Response

Dear Polymers Reviewers,

Re:Resubmission of Manuscript ID: polymers-1597243

Thanks for sending us your comments, which are extremely valuable and helpful for improving our manuscript titled “Energy Storage Application of All-Organic Polymer Dielectrics:A Review”. Thank you very much for giving us another chance to revise the paper. Your valuable suggestions and comments will be very helpful to our manuscript. I really appreciate that you took time out to review our manuscript. All the participating authors have discussed the comments carefully and reached a consensus. We have made the corresponding revisions in the manuscript and highlighted in red. In addition, we have adjusted the layout of the manuscript according to the requirements. After adopting your suggestions, we firmly believe that our manuscript will be greatly improved. Our responses to your comments are listed in the cover letter.

We have further taken this chance to carefully polish the manuscript and resubmitted the revised manuscript to ‘Polymers’.

Thank you for your consideration.  

Sincerely yours,

Yu Feng
